# Online Prediction with Limited Selectivity

**Licheng Liu**
Imperial College London
licheng.liu22@imperial.ac.uk

**Mingda Qiao**
University of Massachusetts Amherst
mqiao@umass.edu

## Abstract

Selective prediction [Dru13, QV19] models the scenario where a forecaster freely decides on the prediction window that their forecast spans. Many data statistics can be predicted to a non-trivial error rate *without any* distributional assumptions or expert advice, yet these results rely on that the forecaster may predict at any time. We introduce a model of Prediction with Limited Selectivity (PLS) where the forecaster can start the prediction only on a subset of the time horizon. We study the optimal prediction error both on an instance-by-instance basis and via an average-case analysis. We introduce a complexity measure that gives instance-dependent bounds on the optimal error. For a randomly-generated PLS instance, these bounds match with high probability.

## 1 Introduction

In *selective prediction* [Dru13, QV19], a forecaster observes $n$ numbers in $[0, 1]$ one by one. At any time $t$, having observed the first $t$ numbers, the forecaster may predict the average of the next $w \leq n - t$ unseen numbers. Both the stopping time $t$ and the window length $w$ are freely chosen by the forecaster, and only one such prediction needs to be made. The goal is to minimize the expected prediction error—the expected squared difference between the forecast and the actual average. How small can this error be?

Surprisingly, Drucker [Dru13] showed that the forecaster can guarantee an error that vanishes as $n \to +\infty$, even though the sequence might be arbitrarily and adversarially chosen. Specifically, this result holds without any distributional assumption on the sequence (other than boundedness), and is thus robust to any misspecification, non-stationarity, or adversarial corruption in the data. Moreover, it directly addresses the prediction error, rather than a regret with respect to a class of experts.

In this paper, we study a variant of the selective prediction model where the forecaster only has *limited selectivity*. Concretely, the forecaster is given a subset of the time horizon, which specifies the timesteps on which they are allowed to make a prediction. This model captures many natural scenarios where predictions are either infeasible or unnecessary during certain time periods. For example, people tend to care about weather forecasts primarily when they have plans for outdoor activities. An investor may be restricted from trading specific commodities during particular seasons, making market predictions less relevant at those times. Similarly, epidemic forecasts are most critical before and during pandemics.

The main contributions of this work are summarized as follows:

- We introduce a theoretical model of Prediction with Limited Selectivity (PLS), which generalizes the selective prediction models of [Dru13, QV19].
- We define a complexity measure termed "approximate uniformity", which gives instance-dependent bounds on the optimal error that a forecasting algorithm can achieve on a PLS instance.
- For PLS instances that are randomly generated according to a $k$-monotone sequence (Definition 4), we show that the instance-dependent bounds match up to a constant factor with high probability.

39th Conference on Neural Information Processing Systems (NeurIPS 2025).

## 1.1 Problem Formulation

**Definition 1** (Prediction with limited selectivity)**.** *The forecaster is given $n$ and a stopping time set $\mathcal{T} \subseteq \{0, 1, 2, \ldots, n-1\}$, and the nature secretly chooses a sequence $x \in [0, 1]^n$. At each timestep $t = 1, 2, \ldots, n$, the forecaster observes $x_t$. At any timestep $t \in \mathcal{T}$, after seeing $x_1, \ldots, x_t$, the forecaster may optionally choose a window length $w \in \{1, 2, \ldots, n-t\}$ and make a prediction $\hat{\mu}$ on the average $\mu = \frac{1}{w} \sum_{i=1}^{w} x_{t+i}$. Once a prediction is made, the game ends and the forecaster incurs an error of $(\hat{\mu} - \mu)^2$. The forecaster must make one prediction before all $n$ numbers are revealed.*

An *instance* of PLS consists of a sequence length $n$ and a stopping time set $\mathcal{T}$. When the forecaster is fully-selective (namely, $\mathcal{T} = \{0, 1, 2, \ldots, n-1\}$), PLS recovers the selective mean prediction setup of [Dru13, QV19]. For simplicity, we focus on the special case of mean prediction, which already captures most of the interesting aspects of PLS. Nevertheless, the problem setup and our results can be easily extended to the more general prediction settings in [QV19]; see Section 5 for more details.

Following prior work on selective prediction, we measure the performance of a forecaster using its *worst-case error* over all possible choices of the sequence $x \in [0, 1]^n$.

**Definition 2.** *The worst-case error of algorithm $\mathcal{A}$ is $\mathrm{error}^{worst}(\mathcal{A}) := \sup_{x \in [0,1]^n} \mathrm{error}(\mathcal{A}, x)$, where $\mathrm{error}(\mathcal{A}, x)$ denotes the expected squared error that $\mathcal{A}$ incurs on sequence $x \in [0, 1]^n$.*

We will frequently use the following equivalent yet more convenient representation for a PLS instance.

**Definition 3.** *The block representation of a PLS instance with sequence length $n$ and stopping time set $\mathcal{T}$ is*

$$\mathcal{L} = (l_1, l_2, \ldots, l_m) = (t_2 - t_1, t_3 - t_2, \ldots, t_m - t_{m-1}, n - t_m),$$

*where $m = |\mathcal{T}|$ and $t_1 < t_2 < \cdots < t_m$ are the elements of $\mathcal{T}$ in ascending order.*

In the rest of the paper, we will use the stopping time set $\mathcal{T}$ and the block representation $\mathcal{L}$ interchangeably to represent a PLS instance. Intuitively, each *block length* $l_i$ corresponds to $l_i$ consecutive timesteps $t_i + 1, t_i + 2, \ldots, t_i + l_i = t_{i+1}$ between adjacent stopping times in $\mathcal{T}$. During these timesteps, the forecaster observes new data but cannot make predictions.

The fully-selective setup of [Dru13, QV19] corresponds to the block representation $\mathcal{L}$ that is an all-one sequence. When $\mathcal{L}$ consists of block lengths of various magnitudes, we naturally expect that the PLS instance becomes harder in the sense that the optimal forecaster has a higher worst-case error. We will formalize this intuition and derive *instance-dependent* error bounds for every PLS instance $\mathcal{L}$ in terms of a simple and combinatorial complexity measure of $\mathcal{L}$.

In addition to the instance-dependent analysis, we will also study settings where the stopping time set $\mathcal{T}$ is randomly generated. Concretely, we consider a setup where each timestep between $0$ and $n-1$ gets included in the stopping time set $\mathcal{T}$ independently, possibly with different probabilities.

**Definition 4** (Random stopping time set)**.** *For integer $n \geq 1$ and $(p_0^\star, p_1^\star, \ldots, p_{n-1}^\star) \in [0, 1]^n$, a $p^\star$-random stopping time set $\mathcal{T}$ is a random subset of $\{0, 1, \ldots, n-1\}$ obtained by independently including each element $t$ with probability $p_t^\star$.*

## 1.2 Our Results

We prove both upper and lower bounds on the optimal worst-case error in PLS, both on an instance-by-instance basis and via an average-case analysis.

**Approximate uniformity.** We introduce a complexity measure, termed *approximate uniformity*, that captures the hardness of a PLS instance.

**Definition 5.** *The approximate uniformity of PLS instance $\mathcal{L} = (l_1, l_2, \ldots, l_m)$ is*

$$\widetilde{U}(\mathcal{L}) := \max_{1 \leq i \leq j \leq m} \frac{l_i + l_{i+1} + \cdots + l_j}{\max\{l_i, l_{i+1}, \ldots, l_j\}}.$$

For the fully-selective case that $\mathcal{L} = (1, 1, \ldots, 1)$, we have $\widetilde{U}(\mathcal{L}) = m$. More generally, $\widetilde{U}(\mathcal{L})$ captures the "effective horizon length" in instance $\mathcal{L}$. As we show in the following, the approximate uniformity roughly characterizes the lowest worst-case error that a forecasting algorithm can achieve.

**Instance-dependent error bounds.** Our main algorithmic contribution is a forecasting algorithm with a worst-case error upper bounded in terms of $\widetilde{U}(\mathcal{L})$. This generalizes the $O(1/\log n)$ error bound of [Dru13] for the fully-selective case.

**Theorem 1.** *For every PLS instance $\mathcal{L}$, there is a forecasting algorithm with a worst-case error of $O(1/\log \widetilde{U}(\mathcal{L}))$.*

We prove Theorem 1 in two steps. First, we establish an $O(1/\log m)$ upper bound for the special case that the $m$ block lengths are *approximately uniform* in the sense of being within a constant factor. Then, we reduce a general PLS instance $\mathcal{L}$ to the special case by merging the blocks into longer blocks with approximately uniform lengths. We show that at least $\Omega(\widetilde{U}(\mathcal{L}))$ longer blocks can be obtained in this way, so the result for the special case implies the $O(1/\log \widetilde{U}(\mathcal{L}))$ upper bound.

Complementary to Theorem 1, we give two lower bounds on the worst-case error.

**Theorem 2.** *For every PLS instance $\mathcal{L} = (l_1, l_2, \ldots, l_m)$, every forecasting algorithm has a worst-case error of $\Omega(\max\{1/[\widetilde{U}(\mathcal{L})]^2, 1/\log m\})$.*

While the first lower bound of $\Omega(1/\widetilde{U}^2)$ does not match the upper bound in Theorem 1, it already has some interesting applications. The following corollary (proved in Appendix A) presents concrete examples where both the sequence length $n$ and the number of blocks $m$ tend to infinity, yet the worst-case error remains lower bounded by a constant. In the first example, the block lengths are geometrically increasing, so $m$ is at most logarithmic in the sequence length $n$. The second example, inspired by the Cantor set, shows that an $\Omega(1)$ error is still unavoidable even if $m$ is polynomial in $n$.

**Corollary 3.** *For every $m \geq 1$, on the PLS instance $\mathcal{L}_m = (2^0, 2^1, 2^2, \ldots, 2^{m-1})$ with sequence length $n = 2^m - 1$ and $m$ blocks, every forecasting algorithm has an $\Omega(1)$ worst-case error. Furthermore, for every $k \geq 1$, there is a PLS instance $\mathcal{L}'_k$ with sequence length $n = 3^k$ and $m = 2^{k+1} - 1$ blocks, on which every forecasting algorithm has an $\Omega(1)$ worst-case error.*

The second lower bound of $\Omega(1/\log m)$ shows that an $m$-block PLS instance is the easiest when all blocks have the same length: An $O(1/\log m)$ worst-case error can be achieved when $l_1 = l_2 = \cdots = l_m$, while no algorithm can achieve a worst-case error $\ll 1/\log m$ on *any* instance with $m$ blocks. While this result might sound intuitive, our proof of the $\Omega(1/\log m)$ bound is non-trivial. The proof involves a novel hierarchical decomposition of the $m$ blocks into a ternary tree and the design of a random process on the resulting tree. This extends a construction of [QV19] for $m$ blocks of equal lengths, which is based on representing the $m$ blocks as a full binary tree of depth $\log m$.

**An average-case analysis.** While the instance-dependent upper and lower bounds do not match on every PLS instance, they *do* match with high probability when the instance is randomly generated. A sequence is *$k$-monotone* if it can be partitioned into at most $k$ contiguous monotone subsequences. Our next result states that, if $\mathcal{T}$ is a $p^\star$-random stopping time set (Definition 4) for a $k$-monotone sequence $p^\star$, both $|\mathcal{T}|$ and $\widetilde{U}(\mathcal{T})$ can be bounded in terms of $\|p^\star\|_1$ with high probability.

**Theorem 4.** *Suppose that $p^\star \in [0,1]^n$ is $k$-monotone and $\mathcal{T}$ is a $p^\star$-random stopping time set. Let $m_0 := \sum_{t=0}^{n-1} p_t^\star$. The following two hold simultaneously with probability $1 - e^{-m_0/3} - 1/n$ over the randomness in $\mathcal{T}$: (1) $|\mathcal{T}| = O(m_0)$; (2) $\widetilde{U}(\mathcal{T}) = \Omega(m_0/(k \log^2 n))$.*

As a direct corollary, our bounds on the optimal worst-case error are tight up to a constant factor with high probability. We prove this corollary in Appendix A.

**Corollary 5.** *If $k$-monotone sequence $p^\star \in [0,1]^n$ satisfies $\sum_{t=0}^{n-1} p_t^\star \geq \Omega((k \log^2 n)^{1+c})$ for some constant $c > 0$, with high probability over the randomness in the $p^\star$-random stopping time set $\mathcal{T}$, the forecasting algorithm from Theorem 1 has a worst-case error that is optimal up to a constant factor.*

Concretely, if $k = O(1)$ is a constant, Corollary 5 applies whenever the expected number of stopping times, $\sum_{t=0}^{n-1} p_t^\star$, is at least $\mathrm{polylog}(n)$. If $k = O(n^\alpha)$ is polynomial in $n$ for some $\alpha \in (0,1)$, we have nearly-tight bounds as long as $\sum_{t=0}^{n-1} p_t^\star = \Omega(n^\beta)$ for some $\beta \in (\alpha, 1]$.

## 1.3 Related Work

Most closely related to our study is the prior work on selective prediction. Drucker [Dru13] introduced the problem of selective mean prediction under the name of "density prediction game" and proved an

$O(1/\log n)$ upper bound on the expected error. Qiao and Valiant [QV19] proved a matching lower bound, and extended the positive result of [Dru13] to the setting of predicting more general functions.

Chen, Valiant, and Valiant [CVV20] introduced a more general framework that encompasses selective prediction as well as other data-collection procedures including importance sampling and snowball sampling. Brown-Cohen [BC21] subsequently obtained a faster algorithm under this framework. Qiao and Valiant [QV21] studied a "learning" variant of selective prediction, in which the learner observes multiple sequences and aims to identify the sequence with the highest average inside a prediction window of their choice.

All the positive results above are based on the Ramsey-theoretic observation that a sufficiently long, bounded sequence must be "predictable" or "repetitive" at *some* scale. This allows a selective forecaster to randomly select a timescale and achieve a vanishing error as the sequence length goes to infinity. Similar observations have been made in different contexts [Fei15, FKT17, MHO25].

More broadly, our setting is related to the recent work on online prediction with abstention, where the forecasting algorithm is allowed to occasionally abstain from making predictions at an additional cost [ZC16, CDG+18, NZ20, GKCS21, GHMS23, PRT+24].

## 2 Instance-Dependent Upper Bounds

### 2.1 Special Case: Approximately Uniform Block Lengths

Recall from Definition 3 that a PLS instance can be represented by a list of block lengths $\mathcal{L} = (l_1, l_2, \ldots, l_m)$. Towards proving Theorem 1, we start with the case that the block lengths do not vary drastically and prove an $O(1/\log m)$ upper bound on the worst-case error. Our algorithm is defined in Algorithm 2. It calls RandomSelect (Algorithm 1) to obtain a randomized prediction position $i$ and length $j$. The algorithm then reads the first $i - 1$ blocks of the sequence, and uses the mean of the last $j$ blocks (namely, blocks $i - j, i - j + 1, \ldots, i - 1$) to predict the mean of the next $j$ blocks ($i$ through $i + j - 1$).

Algorithm 1 (RandomSelect$(s, k)$) is a recursive procedure that prescribes a prediction position and a window length within $2^k$ consecutive blocks (with indices $s$ to $s + 2^k - 1$). With probability $1/k$, it outputs $(s + 2^{k-1}, 2^{k-1})$, i.e., predicting the average of the last $2^{k-1}$ blocks using that of the first $2^{k-1}$. Otherwise, it computes $p$, the proportion of the total length of the first $2^{k-1}$ blocks within the $2^k$ blocks. It then recurses on one of the two halves with probability $p$ and $1 - p$, respectively.

---

**Algorithm 1:** RandomSelect$(s, k)$

**Input:** Integers $s, k \geq 1$.
**Output:** A pair $(i, j)$ such that $s \leq i - j$ and $i + j \leq s + 2^k$.
1 With probability $1/k$, **return** $(s + 2^{k-1}, 2^{k-1})$;
2 $p \leftarrow \left( \sum_{i=0}^{2^{k-1}-1} l_{s+i} \right) / \left( \sum_{i=0}^{2^k-1} l_{s+i} \right)$;
3 **return** RandomSelect$(s, k - 1)$ with probability $p$, and RandomSelect$(s + 2^{k-1}, k - 1)$ with the remaining probability $1 - p$;

---

**Proposition 6.** *On PLS instance $\mathcal{L} = (l_1, l_2, \ldots, l_m)$ that satisfies $\frac{\max\{l_1, l_2, \ldots, l_m\}}{\min\{l_1, l_2, \ldots, l_m\}} \leq C$, Algorithm 2 has a worst-case error of $O(C/\log m)$.*

This extends the result of [Dru13] for $C = 1$, i.e., all blocks have the same length. The proof is similar to those in prior work and deferred to Appendix B. We provide a brief proof sketch below.

*Proof sketch.* For $k \geq 1$ and $\mu \in [0, 1]$, let $L(k, \mu)$ be the maximum squared error that Algorithm 2 incurs on a sequence of $2^k$ blocks with average $\mu$. We prove by induction that $L(k, \mu) \leq O(C/k) \cdot \mu(1 - \mu)$. The proposition then follows from $C/k = O(C/\log m)$ and $\mu(1 - \mu) = O(1)$. □

---

**Algorithm 2:** Prediction with Limited Selectivity on Approximately Uniform Blocks

---
**Input:** Instance $\mathcal{L} = (l_1, l_2, \ldots, l_m)$. Sequential access to sequence $x \in [0,1]^n$ of length
$\qquad n = l_1 + l_2 + \cdots + l_m$.

1  $k \leftarrow \lfloor \log_2 m \rfloor$;
2  $(i, j) \leftarrow \mathsf{RandomSelect}(1, k)$;
3  $t \leftarrow l_1 + l_2 + \cdots + l_{i-1}$;
4  Read $x_1, x_2, \ldots, x_t$;
5  $w_0 \leftarrow l_{i-j} + l_{i-j+1} + \cdots + l_{i-1}$;
6  $\hat{\mu} \leftarrow \frac{1}{w_0}(x_t + x_{t-1} + \cdots + x_{t-w_0+1})$;
7  $w \leftarrow l_i + l_{i+1} + \cdots + l_{i+j-1}$;
8  Predict the mean of $x_{t+1}, \ldots, x_{t+w}$ as $\hat{\mu}$;

---

## 2.2   The General Case

To prove Theorem 1, we merge the $m$ blocks of possibly different lengths into $\approx \widetilde{U}(\mathcal{L})$ blocks with lengths within a constant factor, so that Proposition 6 can be applied to obtain an $O(1/\log \widetilde{U}(\mathcal{L}))$ error bound. Formally, we say that a PLS instance $\mathcal{L}'$ is a *merge* of another instance $\mathcal{L}$, if $\mathcal{L}'$ can be obtained from $\mathcal{L}$ by merging consecutive blocks and taking a contiguous subsequence.

**Definition 6.** $\mathcal{L}' = (l'_1, l'_2, \ldots, l'_{m'})$ *is a merge of* $\mathcal{L} = (l_1, l_2, \ldots, l_m)$ *if there are* $1 \le i_1 < i_2 < \cdots < i_{m'} < i_{m'+1} \le m+1$ *such that* $l'_j = l_{i_j} + l_{i_j+1} + \cdots + l_{i_{j+1}-1}$ *for every* $j \in [m']$.

For instance, $(5,9)$ is a merge of $(1,2,3,4,5,6)$ witnessed by $(i_1, i_2, i_3) = (2,4,6)$: we merge consecutive elements to obtain $(1,5,9,6)$ and take the contiguous subsequence $(5,9)$. Naturally, if the merge of a PLS instance can be solved with a low worst-case error, so can the original instance. We prove the following lemma in Appendix B.

**Lemma 7.** *If* $\mathcal{L}'$ *is a merge of* $\mathcal{L}$, *the minimum worst-case error that can be obtained on* $\mathcal{L}$ *is smaller than or equal to that on* $\mathcal{L}'$.

The next lemma states that any instance $\mathcal{L}$ has a merge of length $\Omega(\widetilde{U}(\mathcal{L}))$ that consists of elements within a constant factor.

**Lemma 8.** *For every* $C > 1$, *every PLS instance* $\mathcal{L}$ *has a merge* $\mathcal{L}' = (l'_1, l'_2, \ldots, l'_{m'})$ *such that: (1)* $m' \ge \lfloor (1 - 1/C) \cdot \widetilde{U}(\mathcal{L}) \rfloor$; *(2)* $\max\{l'_1, l'_2, \ldots, l'_{m'}\}/\min\{l'_1, l'_2, \ldots, l'_{m'}\} \le C$.

*Proof.* By definition of $\widetilde{U}$, there exist indices $1 \le i_0 \le j_0 \le m$ such that $\frac{l_{i_0} + l_{i_0+1} + \cdots + l_{j_0}}{\max\{l_{i_0}, l_{i_0+1}, \ldots, l_{j_0}\}} = \widetilde{U}(\mathcal{L})$. Using the shorthands $L := l_{i_0} + l_{i_0+1} + \cdots + l_{j_0}$ and $M := \max\{l_{i_0}, l_{i_0+1}, \ldots, l_{j_0}\}$, we have $L = \widetilde{U}(\mathcal{L}) \cdot M$. Then, we construct $\mathcal{L}'$ by merging the block lengths $l_{i_0}, l_{i_0+1}, \ldots, l_{j_0}$. We set $T := M/(C-1)$ and greedily form longer blocks of length $\approx T$. Formally, we run the following procedure:

1. Start with $i_1 = i_0$ and counter $\mathrm{cnt} = 1$.

2. Check whether $l_{i_{\mathrm{cnt}}} + l_{i_{\mathrm{cnt}}+1} + \cdots + l_{j_0} < T$. If so, set $m' = \mathrm{cnt} - 1$, $\mathcal{L}' = (l'_1, l'_2, \ldots, l'_{m'})$, and end the procedure.

3. Otherwise, find the smallest $k \in \{i_{\mathrm{cnt}}, i_{\mathrm{cnt}}+1, \ldots, j_0\}$ such that $l_{i_{\mathrm{cnt}}} + l_{i_{\mathrm{cnt}}+1} + \cdots + l_k \ge T$.

4. Set $l'_{\mathrm{cnt}} = l_{i_{\mathrm{cnt}}} + l_{i_{\mathrm{cnt}}+1} + \cdots + l_k$ and $i_{\mathrm{cnt}+1} = k+1$. Increment $\mathrm{cnt}$ by 1 and return to Step 2.

Clearly, the resulting $\mathcal{L}' = (l'_1, l'_2, \ldots, l'_{m'})$ is a merge of $\mathcal{L}$ witnessed by indices $i_1, i_2, \ldots, i_{m'+1}$. The construction ensures that $l'_j \in [T, T + M)$ for every $j \in [m']$. Therefore,

$$\frac{\max\{l'_1, l'_2, \ldots, l'_{m'}\}}{\min\{l'_1, l'_2, \ldots, l'_{m'}\}} < \frac{T + M}{T} = \frac{M/(C-1) + M}{M/(C-1)} = C.$$

The size of the merge is at least $\left\lfloor \frac{L}{T+M} \right\rfloor = \left\lfloor \frac{\widetilde{U}(\mathcal{L}) \cdot M}{M/(C-1)+M} \right\rfloor = \left\lfloor (1 - 1/C) \cdot \widetilde{U}(\mathcal{L}) \right\rfloor$.  $\qquad \square$

Theorem 1 directly follows from Proposition 6, Lemma 7, and Lemma 8.

*Proof of Theorem 1.* Applying Lemma 8 with $C = 2$ shows that $\mathcal{L}$ has a merge $\mathcal{L}' = (l'_1, l'_2, \ldots, l'_{m'})$ such that $m' \geq \lfloor \widetilde{U}(\mathcal{L})/2 \rfloor$ and $\max\{l'_1, l'_2, \ldots, l'_{m'}\}/\min\{l'_1, l'_2, \ldots, l'_{m'}\} \leq 2$. By Proposition 6, there is a forecasting algorithm for $\mathcal{L}'$ with a worst-case error of $O(1/\log m') = O(1/\log \widetilde{U}(\mathcal{L}))$. Then, Lemma 7 gives a forecasting algorithm for $\mathcal{L}$ with the same error bound. $\square$

## 3 Instance-Dependent Lower Bounds

### 3.1 Lower Bound in Terms of Approximate Uniformity

**Theorem 9** (First part of Theorem 2). *For every PLS instance $\mathcal{L}$, every forecasting algorithm $\mathcal{A}$ has a worst-case error of* $\mathrm{error}^{worst}(\mathcal{A}) \geq \frac{1}{16[\widetilde{U}(\mathcal{L})]^2}$.

Recall from Definition 3 that the block representation $\mathcal{L} = (l_1, l_2, \ldots, l_m)$ corresponds to a PLS instance with sequence length $n = l_1 + l_2 + \cdots + l_m$.[1] The $n$ timesteps are naturally divided into $m$ blocks, where each block $i \in [m]$ consists of timesteps $B_i := \{l_1 + l_2 + \cdots + l_{i-1} + j : j \in [l_i]\}$. We prove Theorem 9 using the following observation: Regardless of the prediction window $[t + 1, t + w]$ chosen by the forecasting algorithm, an $\Omega(1/\widetilde{U}(\mathcal{L}))$ fraction of the timesteps within the window must come from the same unseen block.

**Lemma 10.** *Let $\mathcal{L} = (l_1, l_2, \ldots, l_m)$ be a PLS instance with sequence length $n$ and stopping time set $\mathcal{T}$. Then, for every $i_0 \in [m]$, $t = l_1 + l_2 + \cdots + l_{i_0-1} \in \mathcal{T}$ and $w \in [n - t]$, there exists $i \in \{i_0, i_0 + 1, \ldots, m\}$ such that $|B_i \cap [t + 1, t + w]| \geq \frac{w}{2\widetilde{U}(\mathcal{L})}$.*

The proof is deferred to Appendix C. Next, we show how Theorem 9 follows from Lemma 10.

*Proof of Theorem 9 assuming Lemma 10.* Consider the random sequence $x \in \{0, 1\}^n$ constructed by setting all entries within each block to the same bit chosen independently and uniformly at random. Formally, we draw $\mu_1, \mu_2, \ldots, \mu_m \sim \mathsf{Bernoulli}(1/2)$ independently. Then, for each $i \in [m]$ and $j \in B_i$, we set $x_j = \mu_i$.

Fix a forecasting algorithm $\mathcal{A}$. For $t \in \mathcal{T}$ and $w \in [n - t]$, let $\mathcal{E}_{t,w}$ denote the event that $\mathcal{A}$ makes a prediction at time $t$ on $X_{t,w} := \frac{1}{w} \sum_{i=1}^{w} x_{t+i}$. We will show that, conditioning on event $\mathcal{E}_{t,w}$ and any observation $x_{1:t} = (x_1, x_2, \ldots, x_t)$, the conditional variance of $X_{t,w}$ is at least $\Omega(1/[\widetilde{U}(\mathcal{L})]^2)$. This would imply that the conditional expectation of the squared error incurred by $\mathcal{A}$ is lower bounded by $\Omega(1/[\widetilde{U}(\mathcal{L})]^2)$, and the theorem would then follow from the law of total expectation.

Recall that $t \in \mathcal{T}$ must be of form $l_1 + l_2 + \cdots + l_{i_0-1}$ for some $i_0 \in [m]$. Then, $X_{t,w}$ can be equivalently written as $X_{t,w} = \sum_{i=i_0}^{m} \alpha_i \cdot \mu_i$, where $\alpha_i := \frac{1}{w}|B_i \cap [t + 1, t + w]|$ denotes the fraction of timesteps in $[t + 1, t + w]$ that fall into block $B_i$. Since we sample $\mu_1, \ldots, \mu_m$ independently, conditioning on event $\mathcal{E}_{t,w}$ and the observations $x_{1:t}$—both of which are solely determined by the randomness in $\mu_1, \mu_2, \ldots, \mu_{i_0-1}$ and $\mathcal{A}$—each of $\mu_{i_0}, \mu_{i_0+1}, \ldots, \mu_m$ still follows $\mathsf{Bernoulli}(1/2)$ independently and has a variance of $1/4$. Therefore, the conditional variance of $X_{t,w}$ is given by

$$\mathrm{Var}\left[X_{t,w} \mid \mathcal{E}_{t,w}, x_{1:t}\right] = \mathrm{Var}\left[\sum_{i=i_0}^{m} \alpha_i \cdot \mu_i \mid \mathcal{E}_{t,w}, x_{1:t}\right] = \frac{1}{4}\sum_{i=i_0}^{m} \alpha_i^2.$$

By Lemma 10, there exists $i \in \{i_0, i_0 + 1, \ldots, m\}$ such that $\alpha_i \geq \frac{1}{2\widetilde{U}(\mathcal{L})}$, so $\mathrm{Var}\left[X_{t,w} \mid \mathcal{E}_{t,w}, x_{1:t}\right]$ is at least $\frac{1}{4} \cdot \left(\frac{1}{2\widetilde{U}(\mathcal{L})}\right)^2 = \frac{1}{16[\widetilde{U}(\mathcal{L})]^2}$. $\square$

---

[1]Here and in the following, we assume that $0 \in \mathcal{T}$ in the PLS instance. If not, we note that the forecasting algorithm is not allowed to predict until timestep $t_0 := \min \mathcal{T}$, so the analysis goes through after shifting the timesteps by $t_0$.

## 3.2 Hard Instance via Tree Construction

**Theorem 11** (Second part of Theorem 2). *For every PLS instance $\mathcal{L} = (l_1, l_2, \ldots, l_m)$, every forecasting algorithm $\mathcal{A}$ has a worst-case error of $\mathrm{error}^{worst}(\mathcal{A}) \geq \Omega\left(1/\log m\right)$.*

Similar to the proof of Theorem 9, we randomly generate a sequence $x$ by first choosing $\mu \in [0, 1]^m$, and setting every entry within the $i$-th block to $\mu_i$. The key difference is that, instead of drawing each $\mu_i$ independently, we carefully design the correlation between different entries of $\mu$. This correlation structure is specified by a *tree construction* similar to [QV19]: We build a tree whose leaves correspond to the $m$ entries $\mu_1, \ldots, \mu_m$. We assign a noise to each edge of the tree, and the value of a leaf is set to the sum of noises on the root-to-leaf path. Again, we will argue that for every possible prediction window $[t+1, t+w]$, even after seeing the first $t$ entries, the conditional variance in the average of $x_{t+1}, \ldots, x_{t+w}$ is still sufficiently high.

A key difference between our construction and that of [QV19] is that they focused on the special case where $l_1 = l_2 = \cdots = l_m = 1$, so that a full binary tree of depth $\log_2 m$ suffices. In contrast, to handle the general case, our construction involves a ternary tree in which every internal node has either two or three children, depending on whether the current subtree contains a long block whose length dominates the total block length.

We formally introduce our tree construction as follows.

**Definition 7** (Tree construction). *Given a PLS instance $(l_1, l_2, \ldots, l_m)$, we construct a tree using the following recursive procedure:*

- *If $m = 1$, return a tree with a single leaf node that corresponds to $l_1$.*

- *Let $S := l_1 + l_2 + \cdots + l_m$. If there exists $i^\star \in [m]$ such that $l_{i^\star} > S/2$, such index $i^\star$ must be unique, and we construct a ternary tree where: (1) The left subtree of the root node is the tree construction for $(l_1, \ldots, l_{i^\star - 1})$; (2) The middle subtree is a single leaf node corresponding to the $i^\star$-th block; (3) The right subtree is the tree construction for $(l_{i^\star + 1}, \ldots, l_m)$.*

- *Otherwise, every $l_i$ is at most $S/2$. We choose the cutoff $i^\star \in [m]$ as the smallest index such that $l_1 + l_2 + \cdots + l_{i^\star} \geq S/4$. Note that we must have $l_1 + l_2 + \cdots + l_{i^\star} = (l_1 + l_2 + \cdots + l_{i^\star - 1}) + l_{i^\star} \leq S/4 + S/2 = 3S/4$. We recursively build two trees for $(l_1, l_2, \ldots, l_{i^\star})$ and $(l_{i^\star + 1}, l_{i^\star + 2}, \ldots, l_m)$, and return the tree obtained from joining these two subtrees.*

By construction, the tree has $m$ leaf nodes, each of which corresponds to one of the $m$ blocks. For each node $v$ in the tree, let $\mathcal{I}(v)$ denote the set of block indices that correspond to one of the leaves in the subtree rooted at $v$. It is clear that every $\mathcal{I}(v)$ is of form $\{i, i+1, \ldots, j\}$ for some $1 \leq i \leq j \leq m$. We write $\mathrm{size}(v) := |\mathcal{I}(v)|$ as the number of leaves in the subtree rooted at $v$.

Next, we assign a noise magnitude to each node in the tree.

**Definition 8.** *The noise magnitude at node $v$ is set to $\sigma(v) := \sqrt{1 - \frac{\ln(\mathrm{size}(v))}{\ln m}}$.*

The noise magnitude is always in $[0, 1]$: it takes value $0$ at the root node and takes value $1$ at every leaf node. Then, we assign random values in $[0, 1]$ to the nodes in the tree construction as follows.

**Definition 9** (Node value). *The root node $r$ is assigned value $\mu_r = 1/2$ deterministically. Then, for every edge $(u, v)$ in the tree, after $\mu_u$ is determined, we independently draw $\mu_v$ such that:*

$$\mu_v \in \left\{ \frac{1 - \sigma(v)}{2}, \frac{1 + \sigma(v)}{2} \right\} \quad \text{and} \quad \mathbb{E}\left[\mu_v \mid \mu_u\right] = \mu_u.$$

Note that the above is well-defined: By Definition 8, it always holds that $\sigma(u) < \sigma(v)$. So, regardless of whether $\mu_u$ equals $\frac{1 - \sigma(u)}{2}$ or $\frac{1 + \sigma(u)}{2}$, we always have $\mu_u \in \left[\frac{1 - \sigma(v)}{2}, \frac{1 + \sigma(v)}{2}\right]$. Therefore, there exists a unique distribution over $\left\{ \frac{1 - \sigma(v)}{2}, \frac{1 + \sigma(v)}{2} \right\}$ with an expectation of $\mu_u$.

Finally, we construct the hard instance by setting the value of each block to the corresponding node value in the tree construction.

**Definition 10** (Hard instance). *Given a PLS instance $\mathcal{L} = (l_1, l_2, \ldots, l_m)$, we construct a tree following Definition 7 and assign values to its nodes following Definitions 8 and 9. For each $i \in [m]$,*

*let $\mu_i$ denote the value of the leaf node that corresponds to the $i$-th block. Finally, the sequence $x$ consists of $l_1$ copies of $\mu_1$, $l_2$ copies of $\mu_2$, ..., $l_m$ copies of $\mu_m$ in order.*

### 3.3 Structural Properties

Towards proving Theorem 11 using the hard instance from Definition 10, we make some observations on the tree structure as well as the node values. We first note that, for each edge $(u, v)$ in the tree, the conditional variance of $\mu_v$ given $\mu_u$ takes a simple form. Indeed, this is the main motivation behind the choices of the noise magnitudes and node values.

**Lemma 12.** *For every edge $(u, v)$ in the tree and conditioning on any realization of $\mu_u$, it holds that* $\text{Var} \left[ \mu_v \mid \mu_u \right] = \frac{\ln(\text{size}(u)) - \ln(\text{size}(v))}{4 \ln m}$.

*Proof.* Since adding a constant does not change the variance, $\text{Var} \left[ \mu_v \mid \mu_u \right] = \text{Var} \left[ \mu_v - 1/2 \mid \mu_u \right] = \mathbb{E} \left[ (\mu_v - 1/2)^2 \mid \mu_u \right] - \left[ \mathbb{E} \left[ \mu_v - 1/2 \mid \mu_u \right] \right]^2$. By Definition 9, we have $\mathbb{E} \left[ \mu_v \mid \mu_u \right] = \mu_u$, so the second term $\left[ \mathbb{E} \left[ \mu_v - 1/2 \mid \mu_u \right] \right]^2$ reduces to $(\mu_u - 1/2)^2$. Then, we note that $|\mu_u - 1/2| = \sigma(u)/2$ and $|\mu_v - 1/2| = \sigma(v)/2$ always hold, which further simplifies $\text{Var} \left[ \mu_v \mid \mu_u \right]$ into $[\sigma(v)]^2/4 - [\sigma(u)]^2/4$. Finally, the lemma follows from the choices of of $\sigma(u)$ and $\sigma(v)$ in Definition 8. $\square$

The next technical lemma (which we prove in Appendix C) states that, for any $1 \le i \le j \le m$, there exists an edge $(u, v)$ in the tree such that: (1) Observing the first $i - 1$ blocks does not reveal the value of $\mu_v$; (2) The remaining variance of $\mu_v$ has a significant contribution to the average of blocks $i, i + 1, \ldots, j$. To state the lemma succinctly, we define the "total length" of a set $S$. We will mostly use this notation for $S = \mathcal{I}(v)$ (where $v$ is a node in the tree) or $S = [i, j]$ (where $1 \le i \le j \le m$).

**Definition 11.** *The total length of set $S$ is* $\text{totlen}(S) := \sum_{i=1}^{m} l_i \cdot \mathbb{1} \left[ i \in S \right]$.

**Lemma 13.** *For any $1 \le i \le j \le m$, there exists an edge $(u, v)$ in the tree such that: (1) $\mathcal{I}(v) \cap \{1, 2, \ldots, i - 1\} = \emptyset$; (2) $\text{totlen}(\mathcal{I}(v)) \ge \Omega(1) \cdot \text{totlen}([i, j])$; (3) $\text{size}(v) \le \text{size}(u)/2$.*

### 3.4 Lower Bound in Terms of Number of Blocks

We prove Theorem 11 by putting together Lemmas 12 and 13. As in the proof of Theorem 9, we can decompose any prediction window $[t + 1, t + w]$ into several complete blocks $i_0, i_0 + 1, \ldots, j_0 - 1$ and a possibly incomplete block $j_0$. If the complete blocks constitute at least half of the window, we apply Lemma 13 to identify an edge $(u, v)$ in the tree, such that the noise in $\mu_v \mid \mu_u$ contributes an $\Omega(1/\log m)$ variance to the average to be predicted. Otherwise, we lower bound the variance directly by considering the edge above the leaf that corresponds to block $j_0$.

*Proof of Theorem 11.* We consider the random sequence $x \in \{0, 1\}^n$ constructed in Definition 10 and fix a forecasting algorithm $\mathcal{A}$. For $t \in \mathcal{T}$ and $w \in [n - t]$, let $\mathcal{E}_{t,w}$ denote the event that $\mathcal{A}$ makes a prediction at time $t$ on $X_{t,w} := \frac{1}{w} \sum_{i=1}^{w} x_{t+i}$. We will show that, conditioning on any event $\mathcal{E}_{t,w}$ as well as the observations $x_{1:t} = (x_1, x_2, \ldots, x_t)$, the conditional variance of $X_{t,w}$ is at least $\Omega(1/\log m)$. This would lower bound the conditional expectation of the squared error incurred by $\mathcal{A}$ by $\Omega(1/\log m)$, and the theorem would then follow from the law of total expectation.

Recall that $t \in \mathcal{T}$ must be of form $l_1 + l_2 + \cdots + l_{i_0-1}$ for some $i_0 \in [m]$. Let $j_0 \in [m]$ be the smallest number such that $l_{i_0} + l_{i_0+1} + \cdots + l_{j_0} \ge w$. Let $\delta := w - (l_{i_0} + l_{i_0+1} + \cdots + l_{j_0-1})$. Consider the following two cases, depending on whether $\delta$ exceeds half of the window length $w$:

- **Case 1: $\delta \ge w/2$.** Let $v$ be the leaf that corresponds to the $j_0$-th block in the tree construction, and $u$ be the parent of $v$. Note that $\text{size}(v) = 1$ and $\text{size}(u) \ge 2$. By Lemma 12, $\text{Var} \left[ \mu_v \mid \mu_u \right]$ is given by $\frac{\ln(\text{size}(u)) - \ln(\text{size}(v))}{4 \ln m} \ge \frac{\ln 2}{4 \ln m} = \Omega \left( \frac{1}{\log m} \right)$. Furthermore, since event $\mathcal{E}_{t,w}$ and $x_{1:t}$ only depend on the realization of $\mu_1, \mu_2, \ldots, \mu_{i_0-1}$, the value of $\mu_{j_0}$ (namely, $\mu_v$) still has an $\Omega(1/\log m)$ variance conditioning on $\mathcal{E}_{t,w}$ and $x_{1:t}$. Then, since $\delta \ge w/2$ entries among $x_{t+1}, x_{t+2}, \ldots, x_{t+w}$ are set to $\mu_{j_0}$, $\text{Var} \left[ X_{t,w} \mid \mathcal{E}_{t,w}, x_{1:t} \right]$ is at least $\frac{1}{4} \cdot \text{Var} \left[ \mu_{j_0} \mid \mathcal{E}_{t,w}, x_{1:t} \right] \ge \Omega(1/\log m)$.

- **Case 2: $\delta < w/2$.** In this case, we have $\text{totlen}([i_0, j_0 - 1]) = l_{i_0} + l_{i_0+1} + \cdots + l_{j_0-1} = w - \delta \ge w/2$. Applying Lemma 13 with $i = i_0$ and $j = j_0 - 1$ gives an edge $(u, v)$ such

that: (1) $\mathcal{I}(v) \cap \{1, 2, \ldots, i_0 - 1\} = \emptyset$; (2) $\mathrm{totlen}(\mathcal{I}(v)) \geq \Omega(1) \cdot \mathrm{totlen}([i_0, j_0 - 1])$; (3) $\mathrm{size}(v) \leq \mathrm{size}(u)/2$.

By Lemma 12, $\mathrm{Var}\left[\mu_v \mid \mu_u\right]$ is equal to $\frac{\ln(\mathrm{size}(u)) - \ln(\mathrm{size}(v))}{4 \ln m} \geq \frac{\ln 2}{4 \ln m} = \Omega(1/\log m)$. Since event $\mathcal{E}_{t,w}$ and the observation of $x_{1:t}$ only depend on the realization of $\mu_1, \mu_2, \ldots, \mu_{i_0-1}$, and none of these $i_0 - 1$ leaves is inside the subtree rooted at $v$, $\mu_v$ still has an $\Omega(1/\log m)$ conditional variance. Furthermore, within the length-$w$ prediction window, the number of entries that are affected by $\mu_v$ is exactly $\mathrm{totlen}(\mathcal{I}(v)) \geq \Omega(1) \cdot \mathrm{totlen}([i_0, j_0 - 1]) \geq \Omega(1) \cdot w$. Therefore, the conditional variance of $X_{t,w}$ is at least $\Omega(1) \cdot \mathrm{Var}\left[\mu_v \mid \mathcal{E}_{t,w}, x_{1:t}\right] \geq \Omega(1/\log m)$.

$\square$

## 4  An Average-Case Analysis

As a warm-up towards proving Theorem 4, we analyze the special case that $p^\star$ consists of $n$ identical entries. The full proof is presented in Appendix D.

**Proposition 14.** *Let $\mathcal{T}$ be a $p^\star$-random stopping time set for $p^\star = (p, p, \ldots, p) \in [0, 1]^n$. With probability at least $1 - e^{-np/3} - 1/n$ over the randomness in $\mathcal{T}$: (1) $|\mathcal{T}| \leq 2np = O(np)$; (2) $\widetilde{U}(\mathcal{T}) \geq n/\lceil (2 \ln n)/p \rceil - 1 = \Omega(np/\log n)$.*

*Proof.* We first note that $|\mathcal{T}|$ follows $\mathrm{Binomial}(n, p)$, so a multiplicative Chernoff bound gives $\Pr\left[|\mathcal{T}| > 2np\right] \leq e^{-np/3}$. Towards lower bounding $\widetilde{U}(\mathcal{T})$, let $\mathcal{L} = (l_1, l_2, \ldots, l_m)$ be the block representation of $\mathcal{T}$ and let $L_0 := \lceil (2 \ln n)/p \rceil$. By definition, $\widetilde{U}(\mathcal{T})$ is at least $\frac{l_1 + l_2 + \cdots + l_m}{\max\{l_1, l_2, \ldots, l_m\}} = \frac{n - \min \mathcal{T}}{\max\{l_1, l_2, \ldots, l_m\}}$. If we additionally have $\max\{l_1, l_2, \ldots, l_m\} \leq L_0$ and $\min \mathcal{T} \leq L_0$, we would have $\widetilde{U}(\mathcal{T}) \geq \frac{n - L_0}{L_0} = n/\lceil (2 \ln n)/p \rceil - 1$. Thus, it remains to show that, with probability at least $1 - 1/n$, both $\max\{l_1, l_2, \ldots, l_m\} \leq L_0$ and $\min \mathcal{T} \leq L_0$ hold.

Consider the complementary event: for either $\max\{l_1, l_2, \ldots, l_m\} > L_0$ or $\min \mathcal{T} > L_0$ to hold, there must be some $t \in \{0, 1, \ldots, n - L_0 - 1\}$ such that $\mathcal{T} \cap [t + 1, t + L_0] = \emptyset$. For each fixed $t$, $t+1, t+2, \ldots, t+L_0$ get included in $\mathcal{T}$ independently with probability $p$. Thus, $\mathcal{T} \cap [t+1, t+L_0] = \emptyset$ holds with probability $(1-p)^{L_0}$. By the union bound, $\Pr\left[\max\{l_1, l_2, \ldots, l_m\} > L_0 \vee \min \mathcal{T} > L_0\right]$ is at most $(n - L_0) \cdot (1-p)^{L_0}$, which is further upper bounded by $ne^{-pL_0} \leq ne^{-2 \ln n} = 1/n$ using $1 - p \leq e^{-p}$ and $L_0 \geq (2 \ln n)/p$. This completes the proof. $\square$

Our proof of Proposition 14 implies that, when $p^\star$ consists of $n$ copies of the same value $p \in [0, 1]$, the resulting value of $\widetilde{U}$ is at least $\approx np/\log n$ with high probability. Furthermore, the proof of this lower bound *still* holds if each entry of $p^\star$ is *lower bounded by $p$* instead of exactly equal to $p$. To prove Theorem 4, the addition step is then to show that, within each $k$-monotone sequence $p^\star$, we can always find a consecutive subsequence of length $n'$ such that each entry is at least $p'$, and $n'p'$ is at least $\approx \sum_{t=1}^{n} p_t^\star/(k \log n)$. This is done by identifying a monotone subsequence in $p^\star$ that contributes at least a $(1/k)$-fraction of the sum, and then finding an appropriate prefix or suffix of that subsequence.

## 5  Discussion

An obvious open problem is to tighten the instance-dependent error bounds (Theorems 1 and 2). Since the optimal error is $\Theta(1/\log n)$ for the full-selectivity case [Dru13, QV19], one might hope to strengthen the $1/[\widetilde{U}(\mathcal{L})]^2$ lower bound to $1/\log \widetilde{U}(\mathcal{L})$, or at least $1/\mathrm{polylog}(\widetilde{U}(\mathcal{L}))$. Unfortunately, as we show in Proposition 16 (Appendix E), this is not possible: there exists a family of instances $(\mathcal{L}_k)_{k=1}^{+\infty}$ such that $\widetilde{U}(\mathcal{L}_k) \to +\infty$ as $k \to +\infty$, but a worst-case error of $O(1/\widetilde{U}(\mathcal{L}_k))$ can be achieved on each $\mathcal{L}_k$.

Therefore, to obtain tighter instance-dependent bounds, we need to identify a complexity measure that characterizes the hardness of PLS more exactly. A concrete starting point is to examine the instance family from Proposition 16, which is based on a construction that resembles the Cantor set. Roughly speaking, these instances suggest that a sharper complexity measure should account for

the number of approximately uniform blocks that can be obtained via not only merging consecutive blocks, but also "skipping" some shorter blocks at the cost of an additional term in the prediction error.

While we focus on predicting the average of a number sequence, our results can be easily extended to the setting of predicting more general functions (including *smooth* and *concatenation-concave* functions studied by [QV19]). In particular, they imply an $O(1/\log^{1/2} \widetilde{U}(\mathcal{L}))$ upper bound (on the worst-case *absolute* error) for predicting smooth functions and an $O(1/\log \widetilde{U}(\mathcal{L}))$ bound (on the squared error) for concatenation-concave functions. Since the average function is both smooth and concatenation-concave, the lower bounds in Theorem 2 also apply to these broader function classes.

Yet another natural direction is to revisit other classic online learning settings (such as the experts problem) from the perspective of limited selectivity, i.e., when the learner is only allowed to change its prediction or action on some given timesteps. The approximate uniformity measure as well as some of our proof techniques would be natural first steps towards understanding these models.

## Acknowledgments and Disclosure of Funding

We thank the anonymous reviewers, whose suggestions have helped improved this paper.

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

# A Proofs of Corollaries

## A.1 Proof of Corollary 3

**Corollary 3.** *For every $m \geq 1$, on the PLS instance $\mathcal{L}_m = (2^0, 2^1, 2^2, \ldots, 2^{m-1})$ with sequence length $n = 2^m - 1$ and $m$ blocks, every forecasting algorithm has an $\Omega(1)$ worst-case error. Furthermore, for every $k \geq 1$, there is a PLS instance $\mathcal{L}'_k$ with sequence length $n = 3^k$ and $m = 2^{k+1} - 1$ blocks, on which every forecasting algorithm has an $\Omega(1)$ worst-case error.*

*Proof.* In light of the $\Omega(1/[\widetilde{U}(\mathcal{L})]^2)$ lower bound in Theorem 2, it suffices to show that each of these instances has an $O(1)$ approximate uniformity.

**The first family.** Fix $m \geq 1$ and consider $\mathcal{L}_m = (2^0, 2^1, \ldots, 2^{m-1})$. For every $1 \leq i \leq j \leq m$, we have

$$\frac{l_i + l_{i+1} + \cdots + l_j}{\max\{l_i, l_{i+1}, \ldots, l_j\}} = \frac{2^{i-1} + 2^i + \cdots + 2^{j-1}}{2^{j-1}} \leq \frac{2^j}{2^{j-1}} = 2.$$

It follows that $\widetilde{U}(\mathcal{L}_m) \leq 2 = O(1)$.

**The second family.** We construct the instances $\mathcal{L}'_1, \mathcal{L}'_2, \ldots$ recursively: $\mathcal{L}'_1$ is set to $(1, 1, 1)$. For each $k \geq 2$, $\mathcal{L}'_k$ is defined as $\mathcal{L}'_{k-1} \circ (3^{k-1}) \circ \mathcal{L}'_{k-1}$, where $\circ$ denotes sequence concatenation. Then, we prove by induction on $k$ that: (1) each $\mathcal{L}'_k$ corresponds to a PLS instance with $n_k = 3^k$ and $m_k = 2^{k+1} - 1$; (2) $\widetilde{U}(\mathcal{L}'_k) \leq 3$.

For the base case that $k = 1$, we indeed have $n_1 = 3 = 3^k$, $m_1 = 3 = 2^{k+1} - 1$ and $\widetilde{U}(\mathcal{L}'_1) = 3$. For $k \geq 2$, assuming that the statements hold for $\mathcal{L}'_{k-1}$, we have $n_k = n_{k-1} + 3^{k-1} + n_{k-1} = 3^k$ and $m_k = m_{k-1} + 1 + m_{k-1} = (2^k - 1) + 1 + (2^k - 1) = 2^{k+1} - 1$. To upper bound $\widetilde{U}(\mathcal{L}'_k)$, consider an arbitrary contiguous subsequence $(l_i, l_{i+1}, \ldots, l_j)$ in $\mathcal{L}'_k$. If this subsequence contains the entry $3^{k-1}$ in the middle, we have

$$\frac{l_i + l_{i+1} + \cdots + l_j}{\max\{l_i, l_{i+1}, \ldots, l_j\}} \leq \frac{n_k}{3^{k-1}} = \frac{3^k}{3^{k-1}} = 3.$$

If the subsequence does not contain the middle entry, it must be a contiguous subsequence of $\mathcal{L}'_{k-1}$, so the upper bound

$$\frac{l_i + l_{i+1} + \cdots + l_j}{\max\{l_i, l_{i+1}, \ldots, l_j\}} \leq 3$$

would follow from the induction hypothesis. This completes the proof. $\qquad\square$

## A.2 Proof of Corollary 5

Corollary 5 follows from Theorems 1, 2 and 4.

**Corollary 5.** *If $k$-monotone sequence $p^\star \in [0, 1]^n$ satisfies $\sum_{t=0}^{n-1} p_t^\star \geq \Omega((k \log^2 n)^{1+c})$ for some constant $c > 0$, with high probability over the randomness in the $p^\star$-random stopping time set $\mathcal{T}$, the forecasting algorithm from Theorem 1 has a worst-case error that is optimal up to a constant factor.*

*Proof.* Let $m_0 \coloneqq \sum_{t=0}^{n-1} p_t^\star$. Assuming that $m_0 \geq \Omega((k \log^2 n)^{1+c})$, we have $m_0 = \omega(\log n)$ and $k \log^2 n = O(m_0^{1/(1+c)})$. By Theorem 4, it holds with probability $1 - e^{-m_0/3} - 1/n = 1 - O(1/n)$ that $|\mathcal{T}| \leq O(m_0)$ and

$$\widetilde{U}(\mathcal{T}) \geq \Omega\left(\frac{m_0}{k \log^2 n}\right) \geq \Omega\left(\frac{m_0}{m_0^{1/(1+c)}}\right) = \Omega(m_0^{c/(1+c)}).$$

By Theorem 1, there is a forecasting algorithm with a worst-case error of at most

$$O\left(\frac{1}{\log \widetilde{U}(\mathcal{T})}\right) = O\left(\frac{1}{\log\left[m_0^{c/(1+c)}\right]}\right) = O\left(\frac{1}{\log m_0}\right)$$

on the PLS instance $\mathcal{T}$. By Theorem 2, the upper bound $|\mathcal{T}| \leq O(m_0)$ implies that every forecasting algorithm must have an $\Omega(1/\log m_0)$ worst-case error on instance $\mathcal{T}$. This completes the proof. $\quad\square$

## B   Proofs for Section 2

### B.1   Proof of Proposition 6

**Proposition 6.** *On PLS instance $\mathcal{L} = (l_1, l_2, \ldots, l_m)$ that satisfies $\frac{\max\{l_1, l_2, \ldots, l_m\}}{\min\{l_1, l_2, \ldots, l_m\}} \leq C$, Algorithm 2 has a worst-case error of $O(C/\log m)$.*

*Proof.* For integer $k \geq 1$ and $\mu \in [0, 1]$, let $L(k, \mu)$ denote the maximum expected squared loss that Algorithm 2 incurs on a sequence of $2^k$ blocks with average $\mu$. Let $\alpha := \frac{(C+1)^2}{C}$ and $\phi(x) := x(1-x)$. We prove by induction that $L(k, \mu) \leq \frac{\alpha}{k} \cdot \phi(\mu)$. The proposition would then follow from the above together with the observations that $\alpha = O(C)$, $1/k = O(1/\log m)$, and $\phi(\mu) = O(1)$.

**The base case.** When $k = 1$, Algorithm 2 calls $\mathsf{RandomSelect}(1, 1)$, which always returns $(i, j) = (2, 1)$. Then, Algorithm 2 sets

$$t = l_1, \quad w = l_2, \quad w_0 = l_1, \quad \hat{\mu} = \frac{1}{l_1}(x_1 + x_2 + \cdots + x_{l_1}),$$

and predict that the average of $x_{l_1+1}$ through $x_{l_1+l_2}$ is equal to $\hat{\mu}$. Then, the resulting error is given by $(\mu_1 - \mu_2)^2$, where

$$\mu_1 := \frac{1}{l_1}(x_1 + x_2 + \cdots + x_{l_1}) \quad \text{and} \quad \mu_2 := \frac{1}{l_2}(x_{l_1+1} + x_{l_1+2} + \cdots + x_{l_1+l_2})$$

are the averages of the two blocks. Let $c := l_2/l_1$. Note that we have

$$l_1 \cdot \mu_1 + l_2 \cdot \mu_2 = x_1 + x_2 + \cdots + x_{l_1+l_2} = (l_1 + l_2) \cdot \mu.$$

Dividing both sides by $l_1$ gives $\mu_1 + c\mu_2 = (c+1)\mu$.

Then, the squared error can be upper bounded as follows:

$$L(1, \mu) \leq \sup_{\substack{\mu_1, \mu_2 \in [0,1] \\ \mu_1 + c\mu_2 = (c+1)\mu}} (\mu_1 - \mu_2)^2.$$

For any $\mu_1, \mu_2 \in [0, 1]$, $c > 0$ and $\mu = \frac{1}{1+c}\mu_1 + \frac{c}{1+c}\mu_2$, it holds that

$$\phi(\mu) = \frac{1}{1+c}\phi(\mu_1) + \frac{c}{1+c}\phi(\mu_2) + \frac{c}{(1+c)^2}(\mu_1 - \mu_2)^2. \tag{1}$$

Rearranging the above gives

$$(\mu_1 - \mu_2)^2 = \frac{(1+c)^2}{c} \cdot \left[\phi(\mu) - \frac{1}{1+c}\phi(\mu_1) - \frac{c}{1+c}\phi(\mu_2)\right] \leq \frac{(1+c)^2}{c}\phi(\mu),$$

where the second step holds since $\phi(\mu_1)$ and $\phi(\mu_2)$ are non-negative for any $\mu_1, \mu_2 \in [0, 1]$.

Noting that $c \in [1/C, C]$ and that the function $x \mapsto \frac{(x+1)^2}{x}$ is increasing on $[1, +\infty)$ and decreasing on $(0, 1)$, we obtain the base case

$$L(1, \mu) \leq \frac{(c+1)^2}{c} \cdot \phi(\mu) \leq \frac{(C+1)^2}{C} \cdot \phi(\mu) = \frac{\alpha}{1} \cdot \phi(\mu).$$

**Inductive step.** Now, consider the case that $k \geq 2$, assuming that the induction hypothesis holds for $L(k-1, \mu)$. Shorthand $N := 2^{k-1}$ for brevity. When Algorithm 2 calls $\mathsf{RandomSelect}(1, k)$, with probability $1/k$, the pair $(N+1, N)$ is returned, causing Algorithm 2 to predict that the average of blocks $N + 1$ through $2N$ (denoted by $\mu_2$) is the same as the average of the first $N$ blocks (denoted by $\mu_1$). In this case, the squared loss is given by $(\mu_1 - \mu_2)^2$.

With the remaining probability $\frac{k-1}{k}$, RandomSelect recurses on either the first $N$ blocks or the last $N$ blocks. Note that the resulting behavior of Algorithm 2 is exactly identical to when the algorithm runs on either the instance $\mathcal{L}_1 = (l_1, l_2, \ldots, l_N)$ or the instance $\mathcal{L}_2 = (l_{N+1}, l_{N+2}, \ldots, l_{2N})$, which can be controlled using $L(k-1, \cdot)$. Let $c := \frac{\sum_{i=1}^N l_{N+i}}{\sum_{i=1}^N l_i}$. Note that we have $c \in [1/C, C]$, since the assumption that $\frac{\max\{l_1, l_2, \ldots, l_m\}}{\min\{l_1, l_2, \ldots, l_m\}} \leq C$ implies

$$\frac{1}{C} \cdot \sum_{i=1}^N l_i \leq N \cdot \min\{l_1, l_2, \ldots, l_m\} \leq \sum_{i=1}^N l_{N+i} \leq N \cdot \max\{l_1, l_2, \ldots, l_m\} \leq C \cdot \sum_{i=1}^N l_i.$$

Dividing the above by $\sum_{i=1}^N l_i$ gives $1/C \leq c \leq C$. The probability of recursing on the first half is given by $(1 - \frac{1}{k}) \cdot \frac{1}{1+c}$, and that of recursing on the second half is $(1 - \frac{1}{k}) \cdot \frac{c}{1+c}$. Then, by the induction hypothesis, the conditional expectation of the squared loss in this case is at most

$$\frac{1}{1+c} L(k-1, \mu_1) + \frac{c}{1+c} L(k-1, \mu_2) \leq \frac{\alpha}{k-1} \cdot \left( \frac{1}{1+c} \phi(\mu_1) + \frac{c}{1+c} \phi(\mu_2) \right).$$

Then, we get

$$
\begin{aligned}
L(k, \mu) &\leq \sup_{\substack{\mu_1, \mu_2 \in [0,1] \\ \mu_1 + c\mu_2 = (c+1)\mu}} \left[ \frac{1}{k}(\mu_1 - \mu_2)^2 + \frac{k-1}{k} \cdot \frac{\alpha}{k-1} \left( \frac{1}{1+c} \phi(\mu_1) + \frac{c}{1+c} \phi(\mu_2) \right) \right] \\
&\leq \sup_{\substack{\mu_1, \mu_2 \in [0,1] \\ \mu_1 + c\mu_2 = (c+1)\mu}} \left[ \frac{\alpha}{k} \cdot \frac{c}{(c+1)^2}(\mu_1 - \mu_2)^2 + \frac{\alpha}{k} \cdot \left( \frac{1}{1+c} \phi(\mu_1) + \frac{c}{1+c} \phi(\mu_2) \right) \right] \\
&= \frac{\alpha}{k} \cdot \sup_{\substack{\mu_1, \mu_2 \in [0,1] \\ \mu_1 + c\mu_2 = (c+1)\mu}} \left[ \frac{c}{(c+1)^2}(\mu_1 - \mu_2)^2 + \frac{1}{c+1} \phi(\mu_1) + \frac{c}{c+1} \phi(\mu_2) \right] \\
&= \frac{\alpha}{k} \cdot \phi(\mu), \qquad\qquad\qquad\qquad\qquad\qquad\qquad\qquad\qquad\qquad \text{(Equation (1))}
\end{aligned}
$$

where the second step applies $\alpha = \frac{(C+1)^2}{C} \geq \frac{(c+1)^2}{c}$, which follows from $c \in [1/C, C]$. This completes the inductive step and thus the proof. $\qquad\square$

## B.2 Proof of Lemma 7

**Lemma 7.** *If $\mathcal{L}'$ is a merge of $\mathcal{L}$, the minimum worst-case error that can be obtained on $\mathcal{L}$ is smaller than or equal to that on $\mathcal{L}'$.*

*Proof.* Let $\mathcal{L}' = (l'_1, l'_2, \ldots, l'_{m'})$ be a merge of $\mathcal{L} = (l_1, l_2, \ldots, l_m)$ witnessed by indices $i_1, i_2, \ldots, i_{m'+1}$. Let $n := l_1 + l_2 + \cdots + l_m$ and $n' := l'_1 + l'_2 + \cdots + l'_{m'}$ denote the sequence lengths in $\mathcal{L}$ and $\mathcal{L}'$, respectively.

Suppose that a forecasting algorithm $\mathcal{A}'$ has a worst-case error of $\epsilon$ on $\mathcal{L}'$. Then, the following is a forecasting algorithm for $\mathcal{L}$, which we denote by $\mathcal{A}$:

- Let $t_0 := l_1 + l_2 + \cdots + l_{i_1 - 1}$. Read the first $t_0$ elements in the sequence and disregard them.

- Simulate $\mathcal{A}'$. Whenever $\mathcal{A}'$ tries to read the next element in the sequence, read the next element and forward it to $\mathcal{A}'$.

- When $\mathcal{A}'$ predicts $\hat{\mu}$ as the average of the next $w$ elements, make the same prediction.

Clearly, whenever $\mathcal{A}$ runs on sequence $x \in [0,1]^n$, it has the same behavior as $\mathcal{A}'$ running on the length-$n'$ sequence $x' = (x_{t_0+1}, x_{t_0+2}, \ldots, x_{t_0+n'})$. Therefore, the expected error of $\mathcal{A}$ on $x$ is exactly the expected error of $\mathcal{A}'$ on $x'$, which is further upper bounded by $\epsilon$. $\qquad\square$

# C    Proofs for Section 3

## C.1    Proof of Lemma 10

**Lemma 10.** *Let $\mathcal{L} = (l_1, l_2, \ldots, l_m)$ be a PLS instance with sequence length $n$ and stopping time set $\mathcal{T}$. Then, for every $i_0 \in [m]$, $t = l_1 + l_2 + \cdots + l_{i_0-1} \in \mathcal{T}$ and $w \in [n-t]$, there exists $i \in \{i_0, i_0 + 1, \ldots, m\}$ such that $|B_i \cap [t+1, t+w]| \geq \frac{w}{2\widetilde{U}(\mathcal{L})}$.*

*Proof.* Fix $i_0 \in [m]$, $t = l_1 + l_2 + \cdots + l_{i_0-1}$ and $w \in [n-t]$. Recall that the $i$-th block is defined as $B_i := \{l_1 + l_2 + \cdots + l_{i-1} + j : j \in [l_i]\}$. Let $j_0$ be the index of the last block that intersects $[t+1, t+w]$. Formally, $j_0$ is the smallest number in $\{i_0, i_0 + 1, \ldots, m\}$ such that $l_{i_0} + l_{i_0+1} + \cdots + l_{j_0} \geq w$.

Let $\delta := w - (l_{i_0} + l_{i_0+1} + \cdots + l_{j_0-1})$. Note that the prediction window $[t+1, t+w]$ consists of $j_0 - i_0$ complete blocks ($B_{i_0}$ through $B_{j_0-1}$) along with the first $\delta$ timesteps in block $B_{j_0}$. We consider the following two cases, depending on whether $\delta$ exceeds half of the window length $w$:

- **Case 1: $\delta \geq w/2$.** In this case, block $j_0$ would satisfy the lemma. This is because $j_0 \in \{i_0, i_0 + 1, \ldots, m\}$ and

$$|B_{j_0} \cap [t+1, t+w]| = \delta \geq \frac{w}{2} \geq \frac{w}{2\widetilde{U}(\mathcal{L})},$$

  where the last step holds since $\widetilde{U}(\mathcal{L}) \geq 1$ for every $\mathcal{L}$.

- **Case 2: $\delta < w/2$.** In this case, we have $l_{i_0} + l_{i_0+1} + \cdots + l_{j_0-1} = w - \delta \geq w/2$. Furthermore, by definition of $\widetilde{U}(\mathcal{L})$,

$$\widetilde{U}(\mathcal{L}) = \max_{1 \leq i \leq j \leq m} \frac{l_i + l_{i+1} + \cdots + l_j}{\max\{l_i, l_{i+1}, \ldots, l_j\}} \geq \frac{l_{i_0} + l_{i_0+1} + \cdots + l_{j_0-1}}{\max\{l_{i_0}, l_{i_0+1}, \ldots, l_{j_0-1}\}}.$$

  It follows that

$$\max\{l_{i_0}, l_{i_0+1}, \ldots, l_{j_0-1}\} \geq \frac{1}{\widetilde{U}(\mathcal{L})}(l_{i_0} + l_{i_0+1} + \cdots + l_{j_0-1}) \geq \frac{w}{2\widetilde{U}(\mathcal{L})}.$$

  In particular, there exists $i \in \{i_0, i_0 + 1, \ldots, j_0 - 1\}$ such that

$$|B_i \cap [t+1, t+w]| = l_i \geq \frac{w}{2\widetilde{U}(\mathcal{L})}.$$

$\square$

## C.2    Proof of Lemma 13

**Lemma 13.** *For any $1 \leq i \leq j \leq m$, there exists an edge $(u, v)$ in the tree such that: (1) $\mathcal{I}(v) \cap \{1, 2, \ldots, i-1\} = \emptyset$; (2) $\mathrm{totlen}(\mathcal{I}(v)) \geq \Omega(1) \cdot \mathrm{totlen}([i, j])$; (3) $\mathrm{size}(v) \leq \mathrm{size}(u)/2$.*

*Proof.* Let $u_1$ be the deepest node in the tree such that $\{i, i+1, \ldots, j\} \subseteq \mathcal{I}(u_1)$. Note that such a node must exist, since the root node $r$ satisfies $\mathcal{I}(r) = \{1, 2, \ldots, m\} \supseteq \{i, i+1, \ldots, j\}$. Then, we consider two cases, depending on whether $\mathcal{I}(u_1)$ contains a long block that constitutes more than half of the total length.

**Case 1: $\mathcal{I}(u_1)$ contains a long block.** Let $i^\star \in \mathcal{I}(u_1)$ be the unique index such that $l_{i^\star} > \mathrm{totlen}(\mathcal{I}(u_1))/2$. Then, by construction of the tree (Definition 7), $u_1$ has a child $u_2$ that is a leaf node corresponding to the $i^\star$-th block. We claim that $i^\star$ must be in $[i, j]$. Otherwise, by Definition 7, $u_1$ has two other children: one corresponding to blocks $\mathcal{I}(u_1) \cap [1, i^\star - 1]$ and the other corresponding to $\mathcal{I}(u_1) \cap [i^\star + 1, m]$. Then, one of these two children must contain the set $\{i, i+1, \ldots, j\}$, contradicting our choice of $u_1$.

Then, $(u_1, u_2)$ would be the desired edge. For the first condition, since $\mathcal{I}(u_2) = \{i^\star\}$ and $i^\star \geq i$, we have $\mathcal{I}(u_2) \cap \{1, 2, \ldots, i-1\} = \emptyset$. For the second, we have $\mathrm{totlen}(\mathcal{I}(u_2)) = l_{i^\star} > \mathrm{totlen}(\mathcal{I}(u_1))/2 \geq \mathrm{totlen}([i, j])/2$. Finally, since $\mathrm{size}(u_1) \geq 2$ and $\mathrm{size}(u_2) = 1$, it holds that $\mathrm{size}(u_2) \leq \mathrm{size}(u_1)/2$.

**Case 2:** $\mathcal{I}(u_1)$ **has no long blocks.** By Definition 7, $u_1$ has two children $u_2$ and $u_3$ such that $\mathcal{I}(u_1)$ can be partitioned into $\mathcal{I}(u_2)$ and $\mathcal{I}(u_3)$. It follows that $\{i, i+1, \ldots, j\}$ is partitioned into $\mathcal{I}(u_2) \cap [i, j]$ and $\mathcal{I}(u_3) \cap [i, j]$. In particular, we must have

$$\max\{\text{totlen}(\mathcal{I}(u_2) \cap [i, j]), \text{totlen}(\mathcal{I}(u_3) \cap [i, j])\} \geq \text{totlen}([i, j])/2.$$

Without loss of generality, we assume that $\text{totlen}(\mathcal{I}(u_3) \cap [i, j])$ is at least $\text{totlen}([i, j])/2$ in the following; the other case can be handled in a symmetric way.

Let $i'$ be the smallest number in $\mathcal{I}(u_3)$. Then, we have $\mathcal{I}(u_3) \cap [i, j] = \{i', i'+1, \ldots, j\}$. Let $v_1$ denote the deepest node in the subtree rooted at $u_3$ such that $\{i', i'+1, \ldots, j\} \subseteq \mathcal{I}(v_1)$.[2] Then, we further consider the following two subcases, depending on whether $\mathcal{I}(v_1)$ contains a long block (compared to $\text{totlen}(\mathcal{I}(v_1))$):

- **Case 2a:** There exists $i^\star \in \mathcal{I}(v_1)$ such that $l_{i^\star} > \text{totlen}(\mathcal{I}(v_1))/2$. By Definition 7, $v_1$ has a child $v_2$ that is a leaf node corresponding to the $i^\star$-th block. Furthermore, we claim that $i^\star$ must be in $[i', j]$; otherwise, we have $i^\star \geq j + 1$, and $v_1$ has another child with an index set that contains $\{i', i'+1, \ldots, i^\star - 1\} \supseteq \{i', i'+1, \ldots, j\}$, which contradicts our choice of $v_1$.

  Then, $(v_1, v_2)$ would be the desired edge: $\mathcal{I}(v_2) = \{i^\star\}$ and $i^\star \geq i' \geq i$ together give the first condition $\mathcal{I}(v_2) \cap \{1, 2, \ldots, i - 1\} = \emptyset$. For the second condition, we note that $\text{totlen}(\mathcal{I}(v_2)) = l_{i^\star} > \text{totlen}(\mathcal{I}(v_1))/2 \geq \text{totlen}([i', j])/2 \geq \text{totlen}([i, j])/4$. Finally, the third condition follows from $\text{size}(v_2) = 1 \leq \text{size}(v_1)/2$.

- **Case 2b:** Every block in $\mathcal{I}(v_1)$ has a length of at most $\text{totlen}(\mathcal{I}(v_1))/2$. By Definition 7, $v_1$ has a left child $v_2$ that satisfies

$$\text{totlen}(\mathcal{I}(v_2)) \geq \frac{\text{totlen}(\mathcal{I}(v_1))}{4}.$$

Since $\mathcal{I}(v_1)$ contains $\{i', i'+1, \ldots, j\}$, we have

$$\text{totlen}(\mathcal{I}(v_1)) \geq \text{totlen}([i', j]) \geq \frac{\text{totlen}([i, j])}{2}.$$

Combining the above gives $\text{totlen}(\mathcal{I}(v_2)) \geq \text{totlen}([i, j])/8$.

At this point, the edge $(v_1, v_2)$ already satisfies the first two conditions: For the first, we note that $\mathcal{I}(v_2) \subseteq \{i', i'+1, \ldots, j\}$ and is thus disjoint from $\{1, 2, \ldots, i - 1\}$. For the second, we have already shown that $\text{totlen}(\mathcal{I}(v_2)) \geq \text{totlen}([i, j])/8$. However, the last condition $\text{size}(v_2) \leq \text{size}(v_1)/2$ might not hold in general.

Fortunately, this issue can be resolved via yet another case analysis. If $v_2$ is a leaf node, we immediately have the third condition, as $\text{size}(v_2) = 1 \leq \text{size}(v_1)/2$. If there exists $i^\star \in \mathcal{I}(v_2)$ such that $l_{i^\star} > \text{totlen}(\mathcal{I}(v_2))/2$, $v_2$ would have a child $v_3$ that is a leaf corresponding to the $i^\star$-th block. In this case, $(v_2, v_3)$ would be the desired edge, since $\text{totlen}(\mathcal{I}(v_3)) > \text{totlen}(\mathcal{I}(v_2))/2 \geq \text{totlen}([i, j])/16$ and $\text{size}(v_3) = 1 \leq \text{size}(v_2)/2$. Finally, if $\mathcal{I}(v_2)$ does not contain a long block, $v_2$ must have two children $v_3$ and $v_4$ such that

$$\text{size}(v_3) + \text{size}(v_4) = \text{size}(v_2) \quad \text{and} \quad \min\{\text{totlen}(\mathcal{I}(v_3)), \text{totlen}(\mathcal{I}(v_4))\} \geq \frac{\text{totlen}(\mathcal{I}(v_2))}{4}.$$

Without loss of generality, assume that $\text{size}(v_3) \leq \text{size}(v_2)/2$. Then, $(v_2, v_3)$ gives the desired edge, since $\text{totlen}(\mathcal{I}(v_3)) \geq \text{totlen}(\mathcal{I}(v_2))/4 \geq \text{totlen}([i, j])/32$.

$\square$

# D Proof of Theorem 4

We prove Theorem 4 in this appendix. Compared to the proof for constant probability sequences (Proposition 14), our analysis essentially reduces the $k$-monotone case to the constant $p^\star$ case by showing that every $k$-monotone $p^\star$ contains a sufficiently long subsequence, such that the length of the subsequence times the minimum value almost matches $\sum_{t=0}^{n-1} p_t^\star$.

---

[2]Again, such a node must exist, since $u_3$ satisfies $\{i', i'+1, \ldots, j\} \subseteq \mathcal{I}(u_3)$.

**Lemma 15.** *For any $k$-monotone sequence $p^\star = (p_0^\star, p_1^\star, \ldots, p_{n-1}^\star) \in [0,1]^n$, there exists a contiguous subsequence $(p_i^\star, p_{i+1}^\star, \ldots, p_j^\star)$ such that*

$$(j - i + 1) \cdot \min\{p_i^\star, p_{i+1}^\star, \ldots, p_j^\star\} \geq \frac{\sum_{t=0}^{n-1} p_t^\star}{O(k \log n)}.$$

*Proof.* By definition of $k$-monotone sequences, $p^\star$ can be partitioned into at most $k$ contiguous subsequences, each of which is monotone. Therefore, there exist $0 \leq i_0 \leq j_0 \leq n - 1$ such that: (1) $(p_{i_0}^\star, p_{i_0+1}^\star, \ldots, p_{j_0}^\star)$ is monotone; (2) $S := \sum_{t=i_0}^{j_0} p_t^\star \geq \frac{1}{k} \sum_{t=0}^{n-1} p_t^\star$.

Without loss of generality, we assume that $(p_{i_0}^\star, p_{i_0+1}^\star, \ldots, p_{j_0}^\star)$ is non-decreasing; the non-increasing case can be handled symmetrically. We claim that there exists $i \in \{i_0, i_0 + 1, \ldots, j_0\}$ such that

$$(j_0 - i + 1) \cdot p_i^\star \geq \frac{S}{H_n},$$

where $H_n := \frac{1}{1} + \frac{1}{2} + \cdots + \frac{1}{n} = O(\log n)$. Assuming this claim, $(p_i^\star, p_{i+1}^\star, \ldots, p_{j_0}^\star)$ gives the desired subsequence, as it is non-decreasing and satisfies

$$(j_0 - i + 1) \cdot \min\{p_i^\star, p_{i+1}^\star, \ldots, p_{j_0}^\star\} = (j_0 - i + 1) \cdot p_i^\star \geq \frac{S}{H_n} \geq \frac{\sum_{t=0}^{n-1} p_t^\star}{O(k \log n)}.$$

To prove this claim, suppose towards a contradiction that, for every $i \in \{i_0, i_0 + 1, \ldots, j_0\}$, it holds that $(j_0 - i + 1) \cdot p_i^\star < \frac{S}{H_n}$. It then follows that

$$S = \sum_{i=i_0}^{j_0} p_i^\star < \sum_{i=i_0}^{j_0} \left( \frac{S}{H_n} \cdot \frac{1}{j_0 - i + 1} \right) = \frac{S}{H_n} \cdot H_{j_0 - i_0 + 1} \leq S,$$

a contradiction. $\qquad\square$

Finally, we prove Theorem 4 using Lemma 15 and an argument similar to that of Proposition 14.

*Proof of Theorem 4.* We start with the high-probability upper bound on $|\mathcal{T}|$. Let $m_0 := \sum_{t=0}^{n-1} p_t^\star$. Since $|\mathcal{T}|$ is the sum of $n$ independent Bernoulli random variables with means $p_0^\star, p_1^\star, \ldots, p_{n-1}^\star$, we have $\mathbb{E}\left[|\mathcal{T}|\right] = m_0$, and a multiplicative Chernoff bound gives

$$\Pr\left[|\mathcal{T}| > 2m_0\right] \leq e^{-m_0/3}.$$

To lower bound $\widetilde{U}(\mathcal{T})$, we apply Lemma 15 to obtain a contiguous subsequence $(p_{i_0}^\star, p_{i_0+1}^\star, \ldots, p_{j_0}^\star)$ of $p^\star$ such that

$$(j_0 - i_0 + 1) \cdot p_{\min} \geq \frac{\sum_{t=0}^{n-1} p_t^\star}{O(k \log n)} = \frac{m_0}{O(k \log n)},$$

where $p_{\min} := \min\{p_{i_0}^\star, p_{i_0+1}^\star, \ldots, p_{j_0}^\star\}$ denotes the minimum entry in the subsequence. Let $L_0 := \left\lceil \frac{2 \ln n}{p_{\min}} \right\rceil = \frac{O(k \log^2 n)}{m_0} \cdot (j_0 - i_0 + 1)$. Note that we may assume $j_0 - i_0 + 1 \geq 2L_0$ without loss of generality; otherwise, we would have $m_0 = O(k \log^2 n)$, in which case the $\Omega(m_0/(k \log^2 n))$ lower bound on $\widetilde{U}(\mathcal{T})$ would trivially follow from $\widetilde{U}(\mathcal{T}) \geq 1$.

**Good event $\mathcal{E}$.** Next, we define a "good event" which will be shown to imply a lower bound on $\widetilde{U}(\mathcal{T})$ and happen with high probability. For each $t \in \{i_0, i_0 + 1, \ldots, j_0 - L_0 + 1\}$, let $\mathcal{E}_t$ denote the event that $\mathcal{T} \cap [t, t + L_0 - 1] \neq \emptyset$. Let $\mathcal{E} := \bigcap_{t=i_0}^{j_0 - L_0 + 1} \mathcal{E}_t$ be the intersection of these events. In other words, $\mathcal{E}$ is the event that $\{i_0, i_0 + 1, \ldots, j_0\}$ contains no $L_0$ consecutive elements such that none of them is included in $\mathcal{T}$.

**Event $\mathcal{E}$ implies lower bound on $\widetilde{U}$.** When event $\mathcal{E}$ happens, both

$$\mathcal{T} \cap [i_0, i_0 + L_0 - 1] \quad \text{and} \quad \mathcal{T} \cap [j_0 - L_0 + 1, j_0]$$

are non-empty. Thus, if we list the elements in $\mathcal{T} \cap [i_0, j_0]$ in increasing order:

$$t_0 < t_1 < \cdots < t_{m'},$$

we must have $t_0 \le i_0 + L_0 - 1$ and $t_{m'} \ge j_0 - L_0 + 1$. Furthermore, for every neighboring entries $t_{i-1}$ and $t_i$ ($i \in [m']$), we must have $t_i - t_{i-1} \le L_0$; otherwise, $t_{i-1}+1, t_{i-1}+2, \ldots, t_{i-1}+L_0 \in (t_{t-1}, t_i)$ would give $L_0$ consecutive elements that are outside $\mathcal{T}$, contradicting event $\mathcal{E}$.

Let $\mathcal{L} = (l_1, l_2, \ldots, l_m)$ be the block representation of instance $\mathcal{T}$. Note that $\mathcal{L}$ contains a contiguous subsequence

$$(t_1 - t_0, t_2 - t_1, \ldots, t_{m'} - t_{m'-1}).$$

that corresponds to the $m' + 1$ stopping times $(t_0, t_1, \ldots, t_{m'})$. Then, event $\mathcal{E}$ implies that every entry $t_i - t_{i-1}$ is at most $L_0$. Furthermore, the sum of these $m'$ entries is given by

$$\sum_{i=1}^{m'} (t_i - t_{i-1}) = t_{m'} - t_0 \ge (j_0 - L_0 + 1) - (i_0 + L_0 - 1) = (j_0 - i_0 + 1) - 2L_0 + 1.$$

Therefore, by definition of $\widetilde{U}$, $\mathcal{E}$ implies that

$$\widetilde{U}(\mathcal{T}) \ge \frac{\sum_{i=1}^{m'}(t_i - t_{i-1})}{\max\{t_1 - t_0, t_2 - t_1, \ldots, t_{m'} - t_{m'-1}\}} \ge \frac{(j_0 - i_0 + 1) - 2L_0 + 1}{L_0} \ge \frac{j_0 - i_0 + 1}{L_0} - 2.$$

Finally, since $L_0 = \frac{O(k \log^2 n)}{m_0} \cdot (j_0 - i_0 + 1)$, the above gives a lower bound of $\Omega\left(\frac{m_0}{k \log^2 n}\right)$.

**Event $\mathcal{E}$ happens with high probability.** It remains to show that $\Pr[\mathcal{E}] \ge 1 - 1/n$. Fix $t \in \{i_0, i_0 + 1, \ldots, j_0 - L_0 + 1\}$. For each $i \in \{0, 1, \ldots, L_0 - 1\}$, we have $t + i \in [i_0, j_0]$, which implies $p_{t+i}^\star \ge \min\{p_{i_0}^\star, p_{i_0+1}^\star, \ldots, p_{j_0}^\star\} = p_{\min}$. It follows that

$$\Pr\left[\overline{\mathcal{E}_t}\right] = \Pr\left[\mathcal{T} \cap [t, t + L_0 - 1] = \emptyset\right]$$

$$= \prod_{i=0}^{L_0-1} (1 - p_{t+i}^\star) \le \prod_{i=0}^{L_0-1} (1 - p_{\min}) \qquad\qquad (p_{t+i}^\star \ge p_{\min})$$

$$= \exp(-p_{\min} \cdot L_0) \le \frac{1}{n^2}. \qquad\qquad (1 - p \le e^{-p}, L_0 \ge (2 \ln n)/p_{\min})$$

By the union bound, we have

$$\Pr[\mathcal{E}] \ge 1 - \sum_{t=i_0}^{j_0 - L_0 + 1} \Pr\left[\overline{\mathcal{E}_t}\right] \ge 1 - n \cdot \frac{1}{n^2} = 1 - \frac{1}{n}.$$

This completes the proof. $\qquad\qquad\qquad\qquad\qquad\qquad\qquad\qquad\qquad\qquad\qquad\quad\Box$

# E    Instance with $O(1/\widetilde{U}(\mathcal{L}))$ **Worst-Case Error**

**Proposition 16.** *For every $k \ge 2$, there is a PLS instance $\mathcal{L}$ such that: (1) $\widetilde{U}(\mathcal{L}) = 2k$; (2) There is a forecasting algorithm with an $O(1/k)$ worst-case error on $\mathcal{L}$.*

*Proof.* Fix $k \ge 2$. We construct a sequence of instances $\mathcal{L}_1, \mathcal{L}_2, \ldots$ as follows:

- $\mathcal{L}_1 = (1, 1, \ldots, 1)$ is the all-one sequence of length $2k$.

- For every $h \ge 2$, we set

$$\mathcal{L}_h := ((k-1)\mathcal{L}_{h-1}) \circ (2 \cdot (2k)^{h-1}) \circ ((k-1)\mathcal{L}_{h-1}),$$

  where $\circ$ denotes sequence concatenation, and $(k-1)\mathcal{L}_{h-1}$ denotes the sequence obtained from $\mathcal{L}_{j-1}$ by multiplying every entry by $k - 1$.

By a simple induction on $h$, the sum of entries in each $\mathcal{L}_h$ is given by $n_h = (2k)^h$: When $h = 1$, we indeed have $n_1 = 2k = (2k)^1$. For each $h \geq 2$, assuming $n_{h-1} = (2k)^{h-1}$, we have

$$n_h = 2(k-1) \cdot n_{h-1} + 2 \cdot (2k)^{h-1} = (2k)^h.$$

Therefore, for each $h \geq 2$, the middle entry in $\mathcal{L}_h$, $2 \cdot (2k)^{h-1}$, constitutes a $(1/k)$-fraction of the entire sequence length $n_h$. Before and after this middle entry are two copies of $\mathcal{L}_{h-1}$ scaled up by a factor of $k-1$.

In the rest of the proof, we show that $\widetilde{U}(\mathcal{L}_h) = 2k$ for every $h \geq 1$. Furthermore, for all sufficiently large $h$, $\mathcal{L}_h$ admits a forecaster with an $O(1/k)$ worst-case error. These two claims would then prove the proposition.

**Analyze the approximate uniformity.** We start with the direction that $\widetilde{U}(\mathcal{L}_h) \geq 2k$. By construction, the first $2k$ entries of $\mathcal{L}_h$, $(l_1, l_2, \ldots, l_{2k})$, are exactly given by $(k-1)^{h-1}$ times $\mathcal{L}_1$. Thus, the definition of $\widetilde{U}$ gives

$$\widetilde{U}(\mathcal{L}_h) \geq \frac{l_1 + l_2 + \cdots + l_{2k}}{\max\{l_1, l_2, \ldots, l_{2k}\}} = 2k.$$

For the other direction, we show that $\widetilde{U}(\mathcal{L}_h) \leq 2k$ by induction on $h$. When $h = 1$, we clearly have $\widetilde{U}(\mathcal{L}_1) = 2k$. Assuming $\widetilde{U}(\mathcal{L}_{h-1}) \leq 2k$, we analyze $\mathcal{L}_h$. Consider an arbitrary contiguous subsequence $(l_i, l_{i+1}, \ldots, l_j)$ in $\mathcal{L}_h$. If the subsequence contains the middle entry $2 \cdot (2k)^{h-1}$, we would have

$$\frac{l_i + l_{i+1} + \cdots + l_j}{\max\{l_i, l_{i+1}, \ldots, l_j\}} \leq \frac{n_h}{2 \cdot (2k)^{h-1}} = \frac{(2k)^h}{2 \cdot (2k)^{h-1}} = k \leq 2k.$$

Otherwise, $(l_i, l_{i+1}, \ldots, l_j)$ must be a contiguous subsequence of $\mathcal{L}_{h-1}$ scaled up by a factor of $k-1$. It then follows from the induction hypothesis that

$$\frac{l_i + l_{i+1} + \cdots + l_j}{\max\{l_i, l_{i+1}, \ldots, l_j\}} \leq \widetilde{U}(\mathcal{L}_{h-1}) \leq 2k.$$

Therefore, $\widetilde{U}(\mathcal{L}_h) = 2k$ holds for every $h \geq 1$.

**Upper bound the worst-case error.** Next, we give a forecasting algorithm for $\mathcal{L}_h$, which is similar to both the algorithms of [Dru13, QV19] as well as our Algorithm 2:

- If $h = 1$, there are $2k$ blocks of equal length. Predict that the average of the last $k$ blocks is the same as that of the first $k$ blocks.

- If $h > 1$, $\mathcal{L}_h$ consists of three parts: the left half, the middle entry $2 \cdot (2k)^{h-1}$, and the right half. With probability $1/h$, predict that the average of the right half is the same as that of the left half. With the remaining probability $1 - 1/h$, run the same algorithm recursively on either the left or the right half, with equal probability.

For each $h \geq 1$ and $\mu \in [0, 1]$, let $L(h, \mu)$ denote the highest possible squared error that the algorithm above incurs on instance $\mathcal{L}_h$ when the sequence has an average of $\mu$. We will prove by induction that

$$L(h, \mu) \leq \frac{4}{h} \cdot \phi(\mu) + \frac{4}{k},$$

where $\phi(x) = x(1-x)$. It then follows that, for every $h \geq k$, there is a forecasting algorithm for $\mathcal{L}_h$ with a worst-case error of $O(1/k) = O(1/\widetilde{U}(\mathcal{L}_h))$.

**Base case.** For the base case that $h = 1$, let $\mu_1$ and $\mu_2$ be the averages of the two halves, respectively. Clearly, $\mu_1 + \mu_2 = 2\mu$ and the algorithm incurs a squared error of $(\mu_1 - \mu_2)^2$. It follows that

$$L(1, \mu) \leq \sup_{\substack{\mu_1, \mu_2 \in [0, 1] \\ \mu_1 + \mu_2 = 2\mu}} (\mu_1 - \mu_2)^2.$$

We note the identity

$$\phi\left(\frac{a+b}{2}\right) = \frac{\phi(a) + \phi(b)}{2} + \frac{(a-b)^2}{4}, \tag{2}$$

which is a special case of Equation (1) when $c = 1$. Therefore, for any $\mu_1, \mu_2 \in [0,1]$ that satisfy $\mu_1 + \mu_2 = 2\mu$, we have

$$(\mu_1 - \mu_2)^2 = 4\phi\left(\frac{\mu_1 + \mu_2}{2}\right) - 2[\phi(\mu_1) + \phi(\mu_2)] \leq 4\phi(\mu).$$

Thus, we verified the base case that

$$L(1, \mu) \leq 4\phi(\mu) \leq \frac{4}{1} \cdot \phi(\mu) + \frac{4}{k}.$$

**Inductive step.** Consider $h \geq 2$ and assume that the induction hypothesis holds for $L(h-1, \mu)$. Recall that $\mathcal{L}_h$ consists of a left half, a middle block that constitutes a $(1/k)$-fraction of the total length, and a right half. Moreover, the two halves are scaled copies of $\mathcal{L}_{h-1}$. Let $\mu_1$ and $\mu_2$ denote the averages of the two halves, respectively. Let $\mu_0$ denote the average of the middle block. Then, we have

$$\mu = \frac{k-1}{k} \cdot \frac{\mu_1 + \mu_2}{2} + \frac{1}{k} \cdot \mu_0.$$

It follows that

$$\left| \frac{\mu_1 + \mu_2}{2} - \mu \right| = \left| \frac{(\mu_1 + \mu_2)/2 - \mu_0}{k} \right| \leq \frac{1}{k}.$$

Therefore, we have

$$L(h, \mu) \leq \sup_{\substack{\mu_1, \mu_2 \in [0,1] \\ |\mu_1 + \mu_2 - 2\mu| \leq 2/k}} \left[ \frac{(\mu_1 - \mu_2)^2}{h} + \frac{h-1}{h} \cdot \frac{L(h-1, \mu_1) + L(h-1, \mu_2)}{2} \right].$$

Plugging the induction hypothesis $L(h-1, \mu) \leq \frac{4}{h-1} \cdot \phi(\mu) + \frac{4}{k}$ into the above gives

$$L(h, \mu) \leq \sup_{\substack{\mu_1, \mu_2 \in [0,1] \\ |\mu_1 + \mu_2 - 2\mu| \leq 2/k}} \left[ \frac{(\mu_1 - \mu_2)^2}{h} + \frac{4}{h} \cdot \frac{\phi(\mu_1) + \phi(\mu_2)}{2} \right] + \frac{h-1}{h} \cdot \frac{4}{k}.$$

Applying Equation (2) to the supremum above shows that

$$L(h, \mu) \leq \frac{4}{h} \cdot \sup_{\substack{\mu_1, \mu_2 \in [0,1] \\ |\mu_1 + \mu_2 - 2\mu| \leq 2/k}} \phi\left(\frac{\mu_1 + \mu_2}{2}\right) + \frac{h-1}{h} \cdot \frac{4}{k}.$$

Finally, since $\phi(x) = x(1-x)$ is 1-Lipschitz on $[0,1]$, the constraint $|\mu_1 + \mu_2 - 2\mu| \leq 2/k$ implies that $\phi\left(\frac{\mu_1 + \mu_2}{2}\right)$ is $(1/k)$-close to $\phi(\mu)$. Therefore, we conclude that

$$L(h, \mu) \leq \frac{4}{h} \cdot \left[ \phi(\mu) + \frac{1}{k} \right] + \frac{h-1}{h} \cdot \frac{4}{k} = \frac{4}{h} \cdot \phi(\mu) + \frac{4}{k}.$$

This completes the inductive step and thus finishes the proof. $\square$

