# OpenReview forum: "Online Prediction with Limited Selectivity"
_NeurIPS.cc/2025/Conference — NeurIPS 2025 spotlight_

### Official Review · Reviewer_HA5F · 2025-06-23

**Clarity:** 3
**Significance:** 3
**Originality:** 3
**Rating:** 4
**Confidence:** 2

**Summary:**

This paper introduces Prediction with Limited Selectivity (PLS), a variant of selective prediction where forecasters can only make predictions on a predetermined subset of timesteps (rather than any time). The authors develop a complexity measure called "approximate uniformity" and establish instance-dependent upper and lower bounds on the optimal error. They also provide an average-case analysis on k-monotone sequences.

**Questions:**

See Weaknesses.

**Ethical Concerns:**

["NO or VERY MINOR ethics concerns only"]

**Limitations:**

yes

**Quality:**

3

**Strengths And Weaknesses:**

### Strengths

1.  The proposed PLS model addresses realistic constraints where forecasters have limited selectivity, extending selective prediction to practical scenarios. The provided examples (weather forecasting, trading restrictions) are compelling and demonstrate clear real-world relevance.

2. The author provides a clean theoretical framework. The introduced "approximate uniformity" naturally extends the fully-selective to the limited-selective scenarios and captures the hardness of a PLS instance. Using this notation, the paper establishes matching upper and lower bounds.


### Weaknesses

1. The paper is entirely theoretical with lack of empirical evidence. The motivating examples (weather, trading restrictions) are compelling, but there is no simulation demonstrating the algorithm's behavior on synthetic or real sequences. Even small-scale experiments on synthetic or real sequences would help readers gauge convergence speed and constant factors.

2. There is a still large gap between upper ($O(1 / \log (\tilde{U}))$) and lower bounds ($\Omega (1 / \tilde{U}^2)$) on individual instances, which might limit the understanding on worst-case's precision. A short empirical illustration (toy instances displaying the gap) or discussion could strengthen intuition.

---

> ### Author Rebuttal · Authors · 2025-07-28
>
> Thank you for your review! We agree that the empirical evaluation of our methods on synthetic or real sequences as well as the empirical illustration of our error bounds are important directions that would further add to our theory. We restrict the scope of this work to the theoretical aspect of PLS for its conceptual simplicity and the new theoretical insights that it might contribute to selective prediction and online prediction in general. We also note that many prior work on or related to selective prediction (e.g., [Drucker13, Feige15, FKT17, QV19] in Section 1.3) were also purely theoretical.

---

### Official Review · Reviewer_Y8tU · 2025-07-02

**Clarity:** 3
**Significance:** 3
**Originality:** 3
**Rating:** 5
**Confidence:** 3

**Summary:**

This paper introduces a new theoretical model called Prediction with Limited Selectivity (PLS), which is a generalization of the existing selective prediction framework. In this model, a forecaster predicts the average of future values in a sequence but can only initiate a prediction at specific, predetermined times. The authors introduce a complexity measure called "approximate uniformity" to provide instance-dependent upper and lower bounds on the worst-case prediction error. The paper shows that for randomly generated instances, these bounds match with high probability, making them nearly tight.

**Questions:**

Questions for Authors:
1. In the proof of Lemma 8, the procedure for merging blocks seems central to the main result. Could you provide a small, concrete example of this greedy procedure to help solidify the reader's understanding?
2. The ternary tree construction in Definition 7 is a key innovation for the lower bound proof. Is there an intuitive reason why a ternary, as opposed to a binary, structure is necessary to handle blocks of varying lengths? Does it relate to the three cases a sub-problem can fall into (left, right, or dominated by a single large block)?
3. The average-case analysis focuses on k-monotone sequences for the inclusion probabilities p*. What is the motivation behind this specific choice? Does this structure arise in practical scenarios, or is it chosen primarily for its analytical tractability?

**Ethical Concerns:**

["NO or VERY MINOR ethics concerns only"]

**Limitations:**

No serious limitation

**Paper Formatting Concerns:**

No concerns

**Quality:**

3

**Strengths And Weaknesses:**

Strengths:
This is a well-written and theoretically sound paper that addresses a novel and practical extension of the selective prediction problem. The key strengths are:
1. Novel Problem Formulation: The PLS model is a natural and well-motivated generalization of selective prediction. The examples provided, such as weather forecasting for outdoor activities or market predictions during specific trading seasons, effectively illustrate the real-world relevance of limited prediction opportunities.
2. Significant Theoretical Contributions: The introduction of the "approximate uniformity" measure, Ũ(L), is a key contribution that provides an intuitive way to quantify the difficulty of a given PLS instance. The main results (Theorems 1 and 2) provide both upper and lower bounds on the worst-case error in terms of this measure, which is a strong theoretical result.
3. Rigorous Analysis: The proofs are detailed and build upon prior work in a non-trivial way. For instance, the proof of the Ω(1/log m) lower bound involves a novel hierarchical decomposition using a ternary tree, extending the binary tree construction used in previous work.
4. Average-Case Analysis: The paper doesn't stop at worst-case instance analysis. It provides an average-case analysis for randomly generated stopping time sets, showing that the derived bounds are tight with high probability under these conditions. This significantly strengthens the results.

Weaknesses and Areas for Improvement:
While the paper is strong, there are a few areas where it could be improved:

1. Clarity on "Hard" Instances: The paper proves an Ω(1) worst-case error for specific PLS instances, such as those with geometrically increasing block lengths or a Cantor set-like structure. While this is a powerful result, it would be beneficial to provide more intuition as to why these structures are inherently difficult to predict. A brief discussion on how the lack of "uniformity" prevents the forecaster from finding a predictable scale would be helpful.
2. Gap Between Bounds: The authors correctly point out the gap between the upper bound of O(1/log Ũ(L)) and the lower bound of Ω(1/[Ũ(L)]²). While they prove this gap is unavoidable for their complexity measure, it remains the central open question. The discussion in Section 5 is good, but it could be slightly expanded to speculate on what a more exact complexity measure might look like, perhaps drawing more from the properties of the Cantor set-based instances mentioned.

Minor Notational Clarifications: The paper is generally very clear, but some notations could be briefly reiterated. For example, when discussing the proof of Theorem 11, a quick reminder of what µi (the value of the i-th block) represents would aid readability as the reader navigates complex proofs.

---

> ### Author Rebuttal · Authors · 2025-07-28
>
> Thank you for your review!
>
> ---
>
> Regarding your comment on the intuition behind the hard instances, it would be the most informative to follow the proof of Theorem 9 on the instance with geometrically increasing block lengths (i.e., $\mathcal{L} = (1, 2, 4, 8, \ldots)$). Suppose that the nature chooses a random sequence $x$ by setting the bits within each block to either $0$ or $1$ with equal probability, and the choices for different blocks are independent. Then, since the forecaster is only allowed to predict when a block ends, the prediction window must consists of several consecutive blocks, say, $(2^3, 2^4, \ldots, 2^{10})$. Since $2^{10} > 2^3 + 2^4 + \cdots + 2^9$, the last block constitutes more than half of the prediction window. It follows that the randomness of the last block alone contributes an $\Omega(1)$ amount to the variance of the average to be predicted. Therefore, no forecaster can achieve a sub-constant error on this instance.
>
> Regarding your comment on Section 5, our discussion in Appendix E suggests that a sharper complexity measure should not only account for the number of approximately uniform blocks that can be obtained via merging *consecutive* blocks, but also allow "skipping" some shorter blocks at the cost of an additional term in the prediction error. We appreciate your suggestion and will expand this discussion in the revision.
>
> ---
>
> We answer your questions in the following:
>
> 1. Here is a concrete example of the greedy procedure in Lemma 8: Suppose that we are given a block length sequence $L = (2, 1, 3, 4, 1, 1, 5, 1)$, and wish to merge it under a ratio bound $C = 2$.
>
>     The maximum block length is $M = 5$, so the threshold for merging is: $T = M / (C - 1) = 5$. We now run the greedy merging algorithm:
>
>     - Starting at index $1$, we merge the subsequence $(2, 1, 3)$ that has a total length $6 > 5$. We create a merged block of length $l'_1 = 6$.
>
>     - We continue at index $4$: Merging $(4, 1)$ gives a total length of $l'_2 = 5$.
>
>     - Starting at index $6$, we merge $(1, 5)$ to obtain a total length of $l'_3 = 6$.
>
>     - Now we are at index $7$, and the remaining total length (i.e., $1$) is not enough to form a new block of length $\ge 5$, so we stop here.
>
>     The resulting merged sequence is $L' = (6, 5, 6)$, where each merged block length lies in the interval $[T, T + M] = [5, 10]$. The resulting ratio between block lengths also satisfies $(\max l'_j) / (\min l'_j) = 6 / 5 < 2 = C$.
>
> 2. Thank you for this thoughtful observation! Yes, the ternary tree structure in Definition 7 is closely tied to the structure of the lower bound proof. Roughly speaking, the goal of our decomposition is to ensure that, at every split, the total block lengths are divided evenly. When the block lengths are the same (i.e., the setting of [Qiao and Valiant, COLT'19]), doing so is easy. When the lengths can be different, the issue arises when one of the blocks constitutes the vast majority (say, 99.99%) of the total block lengths. In this case, we need to allow a third child that only corresponds to this long block, and use the other two children for the two intervals obtained after removing this long block.
>
> 3. The $k$-monotonicity assumption is a natural way of interpolating between monotone (i.e., $k = 1$) and arbitrary (i.e., $k = n$) sequences. From a practical perspective, it includes all sequences obtained from concatenating $k$ constant sequences, and naturally models scenarios in which the underlying inclusion probability changes periodically (e.g., with a different value for each month / season / year). From a theoretical perspective, being $k$-monotone (sometimes termed "$k$-modal") is a natural assumption in the learning theory literature (e.g., [Daskalakis, Diakonikolas, and Servedio, SODA 2012] and [Canonne, Grigorescu, Guo, Kumar, and Wimmer, Theory of Computing 2019])
>
> ---
>
> [DDS12] Constantinos Daskalakis, Ilias Diakonikolas, and Rocco A. Servedio. Learning k-Modal Distributions via Testing. SODA 2012
>
> [CGGKW19] Clément L. Canonne, Elena Grigorescu, Siyao Guo, Akash Kumar, and Karl Wimmer. Testing $k$-Monotonicity: The Rise and Fall of Boolean Functions. Theory of Computing, 2019

---

### Official Review · Reviewer_2Tme · 2025-07-02

**Clarity:** 3
**Significance:** 3
**Originality:** 3
**Rating:** 4
**Confidence:** 2

**Summary:**

This paper studies the problem of prediction with limited selectivity (PLS), where a forecaster observes a sequence and must choose when and over what window to predict the future mean, using a restricted set of stopping times. The paper introduces approximate uniformity and show that the worst-case prediction error can be bounded in terms of it. The paper provides matching upper and lower bounds up to logarithmic factors, and show that these bounds are tight with high probability when stopping times are randomly generated from a k-monotone distribution.

**Questions:**

1. The algorithm predicts the mean of future blocks by using the mean of past blocks. Is there any assumptions on stationarity of data that supports this? Why should we expect the past data be indicative of the future data?
2. Lemma 8: The lemma only says that there exist a merge that satisfy this, but how could this merge be found?
3. Proof of Lemma 8: T appears to be an tuning parameter to control the max/min ratio of merged block lengths to C. Why is T = M/(C-1) the right choice?
4. Section 4 mentions that this is a warm up, but it would help to highlight what challenges arise when $p^\ast$ is not uniform, and how these uniform-case techniques will generalize.

**Ethical Concerns:**

["NO or VERY MINOR ethics concerns only"]

**Final Justification:**

The paper has audience at NeurIPS and based on my knowledge of the literature, the results are new and may inspire followup works. Thus I think the work deserves acceptance.

**Limitations:**

yes

**Quality:**

3

**Strengths And Weaknesses:**

I am not an expert on online learning with prediction, so I may not have assessed the strength and weakness appropriately. From what I can tell, the results and the analysis techniques are new. The techniques developed in this paper has potentials to be applied elsewhere. The high level idea of connecting locally random structure to approximate uniformity may also be useful in other online learning problems.

The main issues are 1. the algorithm seems to assume access to the block partitioning of the input (i.e., the sequence of block lengths), this is often unrealisitic in applications. 2. Theorem 1 seems to be established under some merge. But it is unclear to me how this merge might be found.

---

> ### Author Rebuttal · Authors · 2025-07-28
>
> Thank you for your review!
>
> ---
>
> Regarding the assumption that the block partitioning (i.e., the stopping time set) is known, we agree that extending our model to scenarios where the stopping times are unknown (e.g., revealed in an online fashion) is a very interesting direction for future work. That said, we believe that this assumption **is** realistic in some scenarios. For example, in our motivating example of weather forecasts, one tends to have a good idea of when they need to travel / commute in the near future. Similarly, in the trading example, the trading windows are typically announced ahead of time.
>
> Regarding your comment on Theorem 1, we find the merge by applying Lemma 8, the proof of which we summarize below. By definition of the approximate uniformity (Definition 5), we can find a consecutive subsequence of blocks with total length $L$ and maximum length $M$ such that $L / M = \widetilde U$. Then, we set parameter $T \coloneqq M / (C - 1)$ and try to merge the blocks greedily from left to right. Whenever the blocks form a longer block of length $\ge T$, we start a new longer block. In this way, we obtain several longer blocks. The length of each longer block is at least $T$, since we never finish a longer block before its length reaches $T$. Each length is at most $T + M$, since the last block added to the longer block is of length $\le M$, i.e., we "overshoot" by at most $M$. This guarantees that the maximum ratio between the lengths of two longer blocks is at most $(T + M) / T = C$ as desired. Furthermore, we form at least $\approx L / (T + M) = (1  -1/C) \cdot (L / M) = (1 - 1/C) \cdot \widetilde U$ longer blocks in this way.
>
> ---
>
> We answer your questions in the following:
>
> 1. No stationarity assumption is needed for this prediction algorithm to be accurate. At a high level, this is due to the Ramsey-type fact that we mentioned on Line 139: a sufficiently long, bounded sequence must be "predictable" or "repetitive" at some scale. This observation is key to the prior results on selective prediction (e.g., [Drucker, TOCT 2013] and [Qiao and Valiant, COLT 2019]).
>
>     As a thought experiment, imagine that the nature is trying to make the sequence as unpredictable as possible for a forecaster that always chooses a window of length $1$. To this end, the nature would draw each number $x_t$ from $\mathsf{Bern}(1/2)$ independently. However, the resulting sequence would be **very** predictable at a larger time scale, e.g., the mean of the entire length-$n$ sequence would be $1/2 \pm O(1/\sqrt{n})$. Conversely, if the nature tries to make the mean of the entire sequence $x$ maximally unpredictable, it would choose $x$ between $(0, 0, \ldots, 0)$ and $(1, 1, \ldots, 1)$ uniformly at random. If this were the case, when the forecaster chooses a short window length $w \ll n$, the average of each length-$w$ window would be exactly that of the past $w$ observations. The takeaway is: the nature cannot choose $x$ in a way such that it is simultaneously unpredictable for all window lengths.
>
> 2. Our proof of Lemma 8 is constructive and algorithmic: the proof gives an explicit algorithm (Lines 186--193) that runs in linear time in the number of blocks and finds the desired merge. (The algorithm is summarized in the above.)
>
> 3. As discussed above, when $T$ is set as the threshold at which we start a new longer block, the length of each longer block is between $T$ and $T + M$, where $M$ is the maximum length of blocks (before merging). Then, the ratio between the lengths of longer blocks would be at most $(T + M) / T = 1 + M / T$. We want this bound to be $C$, so solving $1 + M / T = C$ gives $T = M / (C - 1)$.
>
> 4. Thank you for the comment! We will add the following explanation on the transition from uniform $p^\star$ to general $k$-monotone $p^\star$.
>
>     Our proof of Proposition 14 implies that, when $p^\star$ consists of $n$ copies of $p$, the resulting value of $\widetilde U$ is at least $\approx np / \log n$ with high probability. Furthermore, the proof of this lower bound *still* holds if each entry of $p^\star$ is *lower bounded by* $p$ instead of exactly equal to $p$. To prove Theorem 4, the addition step is then to show that, within each $k$-monotone sequence $p^\star$, we can always find a consecutive subsequence of length $n'$ such that each entry is at least $p'$, and $n'p'$ is at least $\approx \sum_{t=1}^{n}p^{\star}_t / (k\log n)$. This is done by identifying a monotone subsequence in $p^\star$ that contributes at least a $(1/k)$-fraction of the sum, and then finding an appropriate prefix or suffix of that subsequence.

---

> > ### Comment · Reviewer_2Tme · 2025-08-06
> >
> > Thank you for the response and all of my concerns have been addressed. I will maintain my positive rating.

---

### Official Review · Reviewer_KCsM · 2025-07-03

**Clarity:** 4
**Significance:** 3
**Originality:** 4
**Rating:** 5
**Confidence:** 4

**Summary:**

The paper addresses the following problem. In an on-line fashion, $n$ numbers from $[0,1]$ arrive one after the other. The learning agent chooses a moment $t$ to stop and a time horizon $w$ and then tries to guess the average of the next $w$ numbers. No assumption is made about the numbers, so the framework is adversarial.

It is known from the earlier work that this problem has a solution with $o(1)$ loss as $n\to\infty$ (which is quite surprising). This paper deals with the situation when there is a restricted set of possible stopping moments. The result is an upper and lower bound in terms of a combinatorial characteristic of the sequence of stopping moments. If the sequence is exponentially sparse, the worst-case error is $\Omega(1)$.

There is also a study of a probabilistic setup, where every moment can be a stopping moment with some probability. Matching upper and lower bounds are obtained for this case.

**Questions:**

None

**Ethical Concerns:**

["NO or VERY MINOR ethics concerns only"]

**Final Justification:**

Based on the discussion, I would keep my assessment of the paper.

**Limitations:**

Yes

**Quality:**

4

**Strengths And Weaknesses:**

I believe the paper neatly resolves a nice and clearly defined problem.


The problem is not very popular in the community and the literature review is, correspondingly, quite short. The problem, to my knowledge, has no immediate applications. Still I believe the result is nice and warrants publication.

---

> ### Author Rebuttal · Authors · 2025-07-28
>
> Thank you for your review!

---

### Author Response · Authors · 2025-08-09

We thank the AC and reviewers for their valuable time and feedback, which have helped us improve this work. We will revise the content accordingly in the final version, incorporating your suggestions and clarifications.

Best regards,

The authors.

---

### Decision · Program_Chairs · 2025-09-17

**Decision:**

Accept (spotlight)

**Comment:**

This paper considers a novel setting of online prediction with limited selectivity (PLS), wherein there is a pre-defined set of time steps at which the learner is allowed to optionally terminate the game and make a prediction about the average over a number of the next time steps that the learner can decide. This setting generalizes the fully-selective case that was studied earlier and is well motivated by real-world situations. The paper provides the following scientific claims and findings: First, a novel complexity measure that captures the hardness of PLS, in the sense that characterizes both an upper bound and a lower bound of the worst-case instance-dependent prediction error. They also show that their upper and lower bounds match up to a constant factor with high probability when the problem instance is generated randomly according to a $k$-monotone sequence.

**Strengths**: Novel theoretical model of problem for online prediction with limited selectivity, solid theoretical results that give new understanding to PLS via worst-case analysis and average-case analysis. The technical proof ideas and analysis in the paper could potentially helpful for the ML community.

**Weaknesses**: Some concerns about the lack of empirical evaluations and the consideration of a model general model such as unknown stopping times, these are not major concerns and out of scope of the setting that the paper begins with.

Important reasons for accept: See Strengths.

During the review and discussion, all the reviewers appreciate the solid theoretical and conceptual contributions of the paper. All the points raised by the reviewers during the review (mostly minor to my understanding) were well clarified by the authors. The paper easily make a cut.